# A complete toolset for the study of *Ustilago bromivora* and *Brachypodium* sp. as a fungal-temperate grass pathosystem

Franziska Rabe[1,2†], Jason Bosch[1†], Alexandra Stirnberg[1], Tilo Guse[1], Lisa Bauer[1‡], Denise Seitner[1], Fernando A Rabanal[1], Angelika Czedik-Eysenberg[1], Simon Uhse[1], Janos Bindics[1], Bianca Genenncher[1§], Fernando Navarrete[1], Ronny Kellner[3], Heinz Ekker[4], Jochen Kumlehn[5], John P Vogel[6], Sean P Gordon[6], Thierry C Marcel[7], Martin Münsterkötter[8], Mathias C Walter[9¶], Christian MK Sieber[8**], Gertrud Mannhaupt[2,8], Ulrich Güldener[8,9], Regine Kahmann[2], Armin Djamei[1,2*]

[1]Gregor Mendel Institute, Austrian Academy of Sciences, Vienna Biocenter, Vienna, Austria; [2]Max Planck Institute for Terrestrial Microbiology, Marburg, Germany; [3]Max Planck Institute for Plant Breeding Research, Cologne, Germany; [4]Vienna Biocenter Core Facilities GmbH, Vienna, Austria; [5]Leibniz-Institut für Pflanzengenetik und Kulturpflanzenforschung, Gatersleben, Germany; [6]DOE Joint Genome Institute, California, United States; [7]INRA UMR BIOGER, AgroParisTech, Université Paris-Saclay, Thiverval-Grignon, France; [8]Institute of Bioinformatics and Systems Biology, Helmholtz Zentrum München, German Research Center for Environmental Health, Neuherberg, Germany; [9]Department of Genome-oriented Bioinformatics, Wissenschaftszentrum Weihenstephan, Technische Universität München, Freising, Germany

*For correspondence: armin. djamei@gmi.oeaw.ac.at

[†]These authors contributed equally to this work

Present address: [‡]Department of Infectious Diseases and Immunology, Virology Division, Faculty of Veterinary Medicine, University of Utrecht, Utrecht, Netherlands; [§]Max F Perutz Laboratories, Vienna Biocenter, Vienna, Austria; [¶]Bundeswehr Institute of Microbiology, Munich, Germany; [**]Department of Energy, Joint Genome Institute, University of California, Berkeley, United States

Competing interests: The authors declare that no competing interests exist.

**Abstract** Due to their economic relevance, the study of plant pathogen interactions is of importance. However, elucidating these interactions and their underlying molecular mechanisms remains challenging since both host and pathogen need to be fully genetically accessible organisms. Here we present milestones in the establishment of a new biotrophic model pathosystem: *Ustilago bromivora* and *Brachypodium* sp. We provide a complete toolset, including an annotated fungal genome and methods for genetic manipulation of the fungus and its host plant. This toolset will enable researchers to easily study biotrophic interactions at the molecular level on both the pathogen and the host side. Moreover, our research on the fungal life cycle revealed a mating type bias phenomenon. *U. bromivora* harbors a haplo-lethal allele that is linked to one mating type region. As a result, the identified mating type bias strongly promotes inbreeding, which we consider to be a potential speciation driver.

## Introduction

Knowledge of basic molecular principles in biology has mainly been gained through intensive studies of relatively few but technically well accessible model systems ranging from prokaryotes to eukaryotes, including the bacterium *Escherichia coli*, the fungus *Saccharomyces cerevisiae*, the insect *Drosophila melanogaster*, the mammal *Mus musculus*, as well as the dicot plant *Arabidopsis thaliana*. However, findings in these model organisms cannot always be generalized to more distantly related species (*Mohammadi et al., 2015*; *Rine, 2014*). This has led to the development of new model

**eLife digest** Fungi cause many diseases in plants, and reduce the yield of important crops like wheat, corn and rice – all of which belong to the family of grasses. Much research into how disease-causing fungi infect plants will look at a given fungus that infects a specific plant in order to understand plant diseases in general. Over the years, scientists have generated suites of research tools to study these pairs of fungi and plants. However, many of these organism pairs (often called "model pathosystems") have drawbacks when it comes to research in the laboratory, either on the side of the fungus or the side of plant.

*Brachypodium* is a small grass that grows quickly and, unlike crop plants, it grows well in the laboratory. These characteristics make *Brachypodium* a promising model organism for studying many aspects of plant biology. Recently, a fungus called *Ustilago bromivora* – which is related to a fungus that infects corn – was reported to infect *Brachypodium*. This raised the question: could this fungus and this small grass become a new model pathosystem?

Rabe, Bosch et al. set out to answer this question and now provide a toolkit that will help to establish *U. bromivora* and *Brachypodium* as a new model pathosystem. In all of *U. bromivora*'s close relatives, two compatible strains must meet and mate before the fungus can infect the plant; first Rabe, Bosch et al. confirmed that this is also the case for *U. bromivora*. Studying the life cycle of the *U. bromivora* fungus also unexpectedly revealed that while both mating partners are needed to infect the plant, only one of the strains survives outside of the plant after the infection. This phenomenon, referred to as a "mating type bias", has been described for a few other related fungi.

Next, Rabe, Bosch et al. conducted a genetic screen and identified two compatible strains that can grow without the plant as yeast-like cells. This means that these cells can be manipulated genetically, and indeed protocols to grow and genetically engineer the fungus and plant to address different research questions are included in the toolkit as well.

Other new tools include the complete genetic sequence of the fungus with all its genes annotated, and a dataset of which genes are active in *U. bromivora* growing yeast-like in liquid culture versus those active when the fungus grows as a pathogen inside the plant.

Together these new tools and datasets will provide a foundation to study different aspects of the interactions between grasses and disease-causing fungi. This in turn may lead to new methods to reduce fungal growth and reduce yield losses caused by fungal diseases in crop plants. Finally, the discovery that *U. bromivora* shows a mating type bias could provide a starting point for future studies into sexual reproduction in fungi and how new species arise.

systems, closer to species of economic relevance for humans. For temperate poaceous crops like wheat and barley, the small, fast growing grass *Brachypodium distachyon* has become a promising model organism (*Mur et al., 2011*; *Brkljacic et al., 2011*; *Vogel and Hill, 2008*; *Huo et al., 2008*; *Yordem et al., 2011*; *Garvin, 2008*; *Draper et al., 2001*). The intrinsic properties of *B. distachyon*, including self-fertility, its short life cycle, a small sequenced diploid genome, and its genetic accessibility, make it highly suitable as a laboratory model plant. Symbiotic and biotrophic interactions of plants with fungi or other microbes can have major impacts on plant development and crop yield (*Oerke, 2006*). Therefore, the study of these interactions is important for research-driven pest control. Biotrophic plant pathogens rely on a living host to proliferate and complete their life cycles. To successfully colonize their host plants, these pathogens employ small secreted molecules, termed effectors. By means of these molecules, biotrophic pathogens are able to avoid host recognition, suppress plant defense responses, and redirect the host metabolism for their needs (*Djamei et al., 2011*; *Doehlemann and Hemetsberger, 2013*; *Djamei and Kahmann, 2012*; *Deslandes and Rivas, 2012*; *Bozkurt et al., 2012*). Since most effectors target processes on the host side, the understanding of biotrophy and the molecular study of effectors profits strongly from both, a fully genetically accessible host plant as well as a pathogen that is amenable to genetic manipulation.

Among the facultative biotrophic pathogens, the smut fungus *Ustilago maydis* is a valuable model to study biotrophic interactions (*Djamei et al., 2011*; *Brefort et al., 2014*; *Kämper et al., 2006*; *Horst et al., 2010*; *Brefort et al., 2009*; *Doehlemann et al., 2009*). Although *U. maydis* is

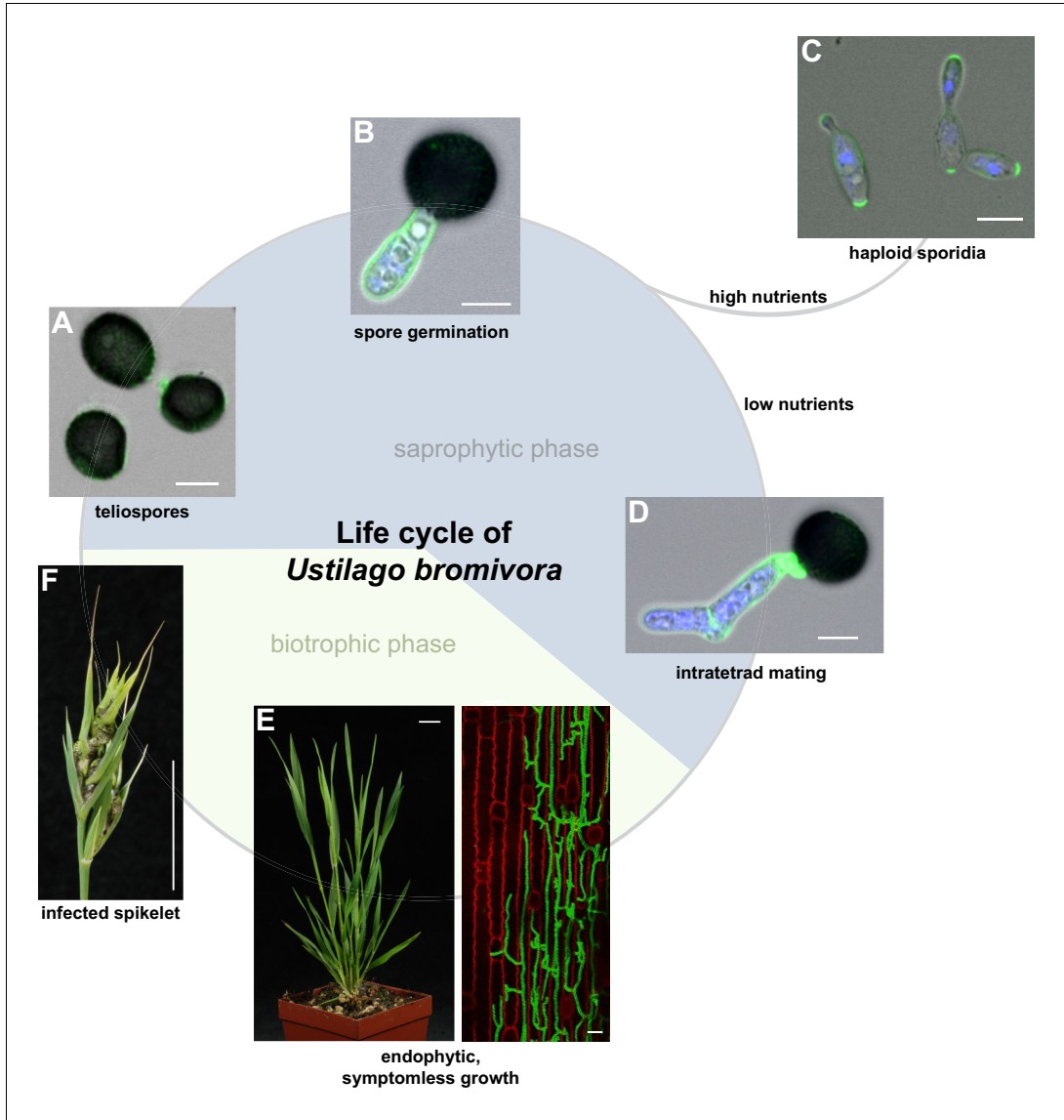

**Figure 1.** The life cycle of *Ustilago bromivora*. *U. bromivora* spores germinate (**A**) and form a promycelium (**B**). Under high nutrient conditions, haploid yeast-like progeny (sporidia) are released and proliferate via budding (**C**). Under low nutrient conditions intratetrad mating occurs between two adjacent cells of a promycelium by formation of a loop-like mating structure that connects both cells (**D**). After plant penetration, fungal filaments grow mainly along the stem without triggering macroscopic symptoms (**E**) until flower development occurs. Upon flowering, macroscopic symptoms are detectable as black, smutted spikelets filled with fungal spores (**F**). Fungal cell walls and nuclei were stained with WGA-Alexa Fluor 488 and DAPI, respectively (**A–D**). Plant membranes were stained with FM4-64, fungal hyphae with WGA-Alexa Fluor 488 (**E**). Scale bars: 5 μm (**A–D**), 10 μm (**E**, right panel), or 1 cm (**E**, left panel) and (**F**).

The following figure supplement is available for figure 1:

**Figure supplement 1.** The morphology of *U. bromivora* sporidia is similar to *U. hordei*, but not to *U. maydis*.

genetically fully accessible with a small, completely sequenced genome of 20.5 Mb and many molecular tools available, its host plant *Zea mays* is not as suitable as laboratory model organism due to its large size, demanding growth requirements, cross-pollinating nature, elaborate transformation requirements, and complex genome (*Kämper et al., 2006*; *Que et al., 2014*; *Schnable et al., 2009*). Accordingly, a more suitable model system where both, plant and pathogen are more

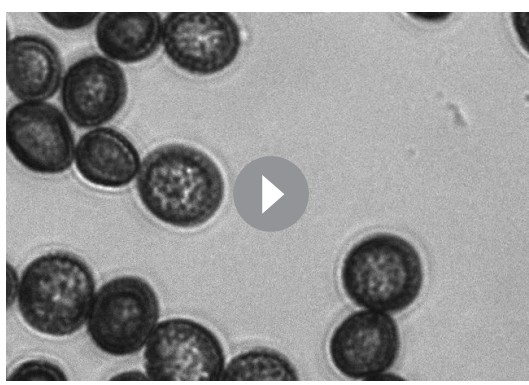

**Video 1.** Spore germination under high nutrient conditions (PD agar).

accessible is highly desirable. The recent rediscovery that the *U. maydis*-relative *U. bromivora* can infect *Brachypodium* sp. provided the impetus to explore the suitability of the *U. bromivora-Brachypodium* system as a novel plant-pathogen model to study biotrophic interactions (*Barbieri et al., 2011*).

Here we describe the characterization of this valuable model pathosystem. This encompasses (i) the detailed understanding of the life cycle *of U. bromivora*, (ii) the identification of a sequenced compatible host, (iii) the establishment of transformation systems for both the pathogen and the plant, and (iv) the sequencing and analysis of the fungal genome. Moreover, the observation of a mating type bias present in *U. bromivora* provided first insights into the biology of this pathogen.

## Results and Discussion

### The life cycle of *U. bromivora* and its mating type bias

Descriptive studies on the life cycle and mating behavior of smut fungi have been performed for more than 100 years (*Bauch, 1922*; *Brefeld, 1883*). In all grass smuts (family Ustilaginaceae) studied so far, sexual and pathogenic development are coupled (*Begerow et al., 2014*). Infection mostly occurs at the early seedling stage of the respective host plant. After penetrating the plant, the dikaryotic pathogen grows biotrophically inter- and intracellularly in its host and macroscopic symptoms are usually exclusively limited to the inflorescences (*Brefort et al., 2009*). In these plant organs, fungal proliferation finally occurs after systemic growth and eponymous black teliospores are

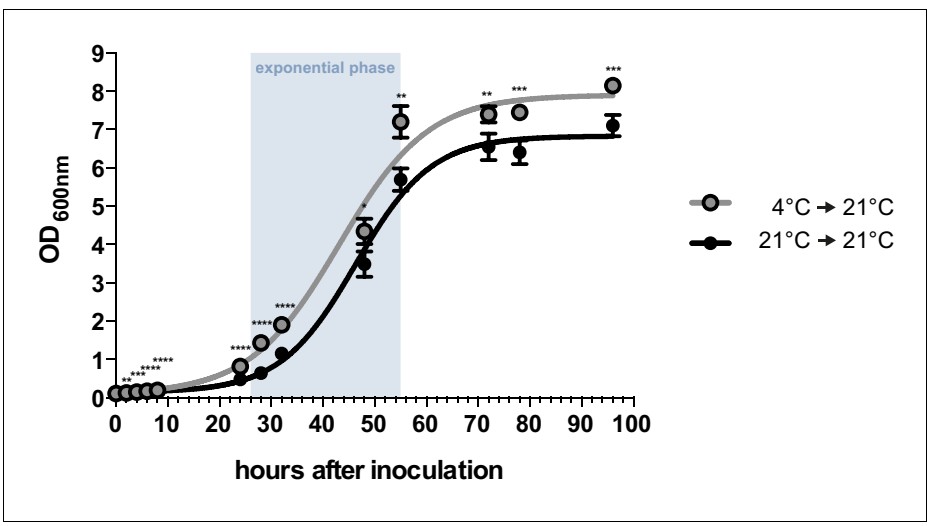

**Figure 2.** Axenic growth of *U. bromivora*. Growth of *U. bromivora* UB1 was assessed by monitoring cell density spectrophotometrically at $\lambda = 600$ nm in liquid PD medium at 21°C in a time course of 96 hr. Growth was compared between axenic cultures inoculated from plate with cold-treated fungal cell material (24 hr at 4°C) or cell material that was kept at 21°C. Experiments were performed in 3 biological replicates. Significance between cell densities of cold-treated and non-treated cells at each time point was calculated by unpaired *t* test. **$p<0.01$, ***$p<0.001$, ****$p<0.0001$. The doubling time was calculated by taking the slope of a linear regression during exponential phase.

formed. *U. maydis* is a prominent exception as it induces spore filled galls on all aerial parts of its host (*Brefort et al., 2009*). As described in the early 20th century by Robert Bauch (*Bauch, 1925*), *U. bromivora* grows in its host plant after infection at the seedling stage, and only exhibits macroscopic symptoms during inflorescence development by replacing flowers with black teliospore-filled sori (*Figure 1*). Teliospores are the diploid resting stage of these fungi and are able to survive harsh environmental conditions. In a humid, favorable environment, spores germinate, undergo meiosis and form a promycelium from which haploid cells are released. The haploid yeast-like cells are non-pathogenic and grow saprophytically (*Brefort et al., 2009*). To elucidate this part of the life cycle of *U. bromivora*, we followed germination of fungal spores by widefield and confocal laser scanning microscopy. Under nutrient-rich conditions, *U. bromivora* spores germinated, formed a promycelium and yeast-like cells were released (*Figure 1A–C*, *Video 1*). The egg-shaped cells of *U. bromivora*, which are released after germination, show high morphological similarity to the sporidia of the barley-infecting covered smut *Ustilago hordei,* and their morphology differ from the cigar-like shaped haploid cells of *U. maydis* (*Figure 1—figure supplement 1*). The high morphological similarity between *U. bromivora* and *U. hordei* is in line with phylogenetic analysis based on internal transcribed spacer (ITS) and large subunit (LSU) rDNA sequence comparison, which suggested that *U. bromivora* is more closely related to *U. hordei* than to *U. maydis* (*Stoll et al., 2005*).

During saprophytic growth, daughter cells bud off from the mother cell at the side of the oval tip. For a better visualization of the fungal cells, we employed the chitin stain Wheat Germ Agglutinin (WGA) that is conjugated to Alexa Fluor 488 (*Figure 1C*). The WGA-Alexa Fluor 488 conjugate distinctively stained the tip of the sporidia indicating cell wall composition differences at the sporidial

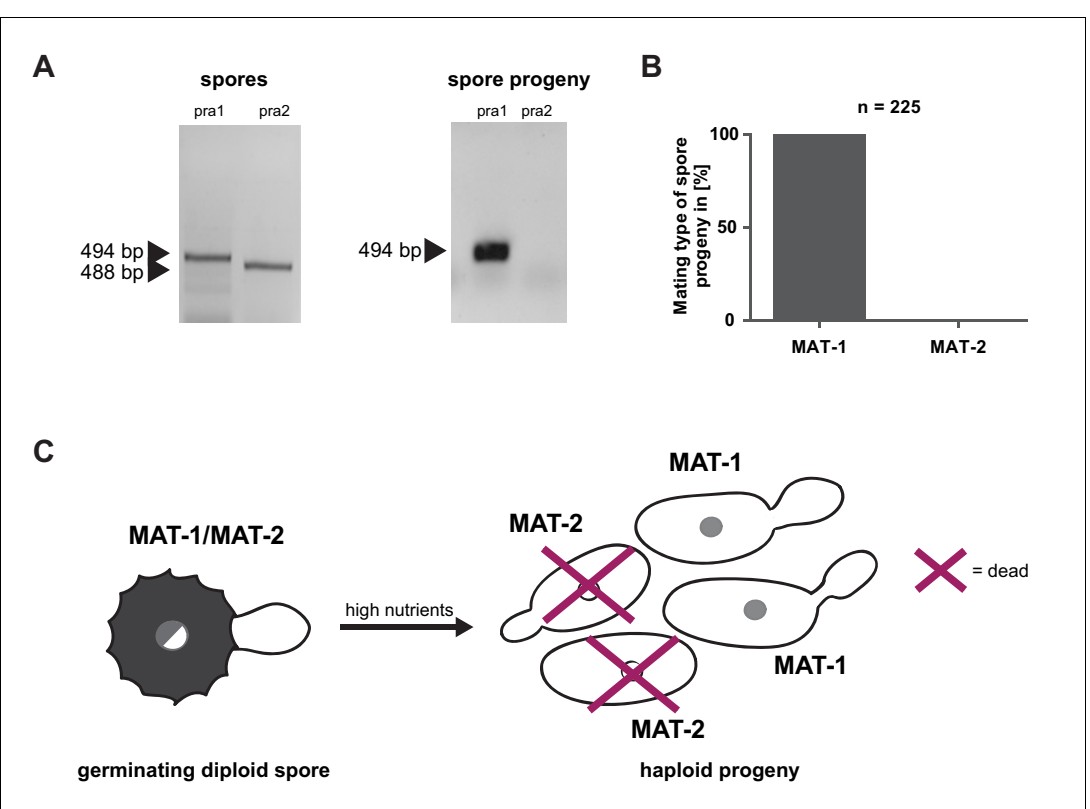

**Figure 3.** Mating type bias of *U. bromivora*. (**A**) Diagnostic PCR on genomic DNA derived from spores and spore progeny to test for mating type 1 (MAT-1) or mating type 2 (MAT-2). To this end, primers targeting a conserved region of pheromone receptor alleles 1 (*pra1*) and 2 (*pra2*), adapted from *Kellner et al. (2011)*, were used. Sizes of PCR products are indicated with arrow heads. Representative PCR results are shown. (**B**) Quantification of mating type alleles of 225 progeny derived from 21 spores by PCR as described in (**A**). (**C**) Schematic model illustrating the observed mating type bias phenomenon.

poles. This could be due to either an uneven chitin distribution or an uneven accessibility of chitin for the stain along the sporidial cell wall (*Figure 1C*).

By performing growth analyses we determined the doubling time of *U. bromivora* sporidia. In comparison to *U. maydis* sporidia that exhibit a doubling time of about 2 hr (*Steinberg, 2007*), *U. bromivora* sporidia grow much slower (doubling time in exponential phase at 21°C: 5 1/3 hr; *Figure 2*). Interestingly, we observed that the lag-phase of the *U. bromivora* culture could be significantly shortened when *U. bromivora* was kept for 24 hr at 4°C prior to inoculation (*Figure 2*). The underlying mechanism for this phenomenon is unclear and awaits further research.

Prior to pathogenic growth, in case of the well-studied smut fungus *U. maydis,* two haploid cells of compatible mating type recognize each

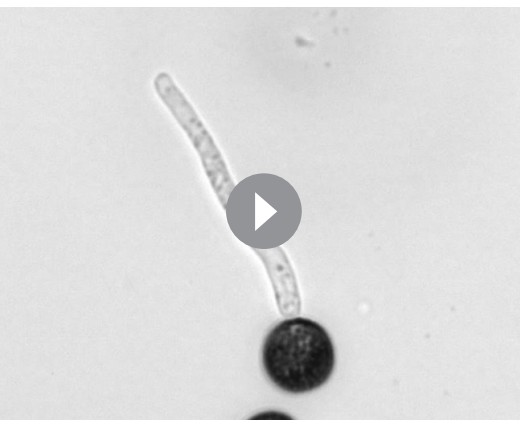

**Video 2.** Spore germination under low nutrient conditions (water agar).

other via a pheromone-receptor system on the plant surface, grow towards each other, fuse, and form a dikaryotic filament. This dikaryotic filament represents the pathogenic form of the fungus. It penetrates the leaf cuticle and grows intra- and intercellularly inside the maize plant. After proliferation in the aerial parts of the host, the filaments fragment and, after karyogamy, form diploid spores (*Brefort et al., 2009*; *Martinez-Espinoza et al., 2002*). The molecular regulation of the dimorphic switch is based on the bi-allelic *a* locus encoding the pheromone receptor system and the multiallelic *b* locus coding for a heterodimeric homeodomain transcription factor that controls pathogenic development (*Schulz et al., 1990*; *Bölker et al., 1992*). While *U. maydis* has a tetrapolar mating type system with *a* and *b* loci that segregate during meiosis, these two loci are genetically and physically linked in *U. hordei*, leading to a bipolar mating system with only two mating types, mating type 1 (MAT-1) and mating type 2 (MAT-2) (*Bakkeren and Kronstad, 1994*; *Lee et al., 1999*). To test for the presence of the mating system described for related smuts we assessed whether *U. bromivora* spores, that are considered to contain the genetic information of both mating partners, harbor known pheromone-receptor alleles. To this end, we employed a diagnostic PCR approach described by *Kellner et al. (2011)* using primers that target conserved regions of either the pheromone receptor allele *pra1* or *pra2* -. The PCR analysis revealed amplicons for *pra1* and *pra2* indicating the presence of both mating type alleles (*Figure 3A*). To subsequently identify haploid *U. bromivora* cells of compatible mating type after spore germination, 225 randomly chosen progeny of 21 spores were tested. Surprisingly, by testing these progeny, we identified only cells of mating type 1 (MAT-1; *Figure 3A,B*). These findings indicate a mating type bias after spore germination (*Figure 3C*). This phenomenon of biased strains was previously described in other related fungi where recessive alleles linked to the mating type-locus were found to be causative for the observed biases (*Nielsen, 1968*; *Hood and Antonovics, 2000*).

Due to the putative haplo-lethal allele, we assumed that haploid MAT-2 cells have only a very short period of time to be rescued by finding and mating with a compatible partner, allowing the subsequent formation of a dikaryotic filament. To observe mating events, spores were germinated on water agar and monitored by light microscopy (*Video 2*, *Figure 1D*). After spore germination, the cytoplasmic connection to the spore content becomes separated and, in the majority of germination events, only two cells are visible outside the spores. These cells frequently form a cytoplasmic bridge-like mating structure and fuse (*Video 2*, *Figure 1D*). The mating of progeny derived from the same spore is also known as an intratetrad mating event (*Antonovics and Abrams, 2004*). Using DAPI staining at this stage, we most commonly observed a diffuse distribution of signal across the two mating cells and, more rarely, a distinct nuclei-like area (*Figure 1D*). In contrast to these mating cells, distinct nuclei-like structures could be visualized with this stain in saprophytically growing cells (*Figure 1C*). The destiny of the other two meiotic products, which, in the majority of observed cases, seemed to stay inside the spore shell, is unclear and awaits further research. The formation of conjugation hyphae that loop directly to neighboring, conjoined cells of the same tetrad could be a

**Table 1.** Tested *Brachypodium* accessions for *U. bromivora* susceptibility/resistance.

| Name | Species | Sequenced | Susceptible to *U. bromivora* | Accession number | Source | Country of origin |
|---|---|---|---|---|---|---|
| ABR3 | *B. distachyon* | Y | N (2x)* | ABY-Bs 5088 | Brachyomics collections (C. Stace and P. Catalán), Aberystwyth, UK | Spain |
| ABR4 | *B. distachyon* | Y | Y (many) | ABY-Bs 5089 | Brachyomics collections (C. Stace and P. Catalán), Aberystwyth, UK | Spain |
| ABR6 | *B. distachyon* | Y | N (2x)* | ABY-Bs 5091 | Brachyomics collections (C. Stace and P. Catalán), Aberystwyth, UK | Spain |
| ABR7 | *B. distachyon* | Y | N (2x)* | ABY-Bs 5092 | Brachyomics collections (C. Stace and P. Catalán), Aberystwyth, UK | Spain |
| ABR9 | *B. distachyon* | Y | N | - | unknown | Croatia |
| Adi-10 | *B. distachyon* | Y | N | W6 39243 | USDA-ARS-WRPIS; *Vogel et al. (2009)* | Turkey |
| Adi-12 | *B. distachyon* | Y | N | W6 39245 | USDA-ARS-WRPIS; *Vogel et al. (2009)* | Turkey |
| Adi-2 | *B. distachyon* | Y | N | W6 39235 | USDA-ARS-WRPIS; *Vogel et al. (2009)* | Turkey |
| Bd1-1 | *B. distachyon* | Y | Y | PI 170218 / W6 46201 | GRIN Germplasm; *Vogel et al. (2006)* | Turkey |
| Bd18-1 | *B. distachyon* | Y | N | PI 245730 / W6 46204 | USDA-ARS-WRPIS; *Vogel et al. (2006)* | Turkey |
| Bd21 | *B. distachyon* | Y | N | PI 254867 / W6 36678 | GRIN Germplasm; *Vogel et al. (2006)* | Iraq |
| Bd21-3 | *B. distachyon* | Y | N | W6 39233 | GRIN Germplasm; *Vogel and Hill (2008)* | Iraq |
| Bd2-3 | *B. distachyon* | Y | N | PI 185133 / W6 46202 | *Vogel et al. (2006)* | Iraq |
| Bd3-1 | *B. distachyon* | Y | N (2x)* | PI 185134 / W6 46203 | USDA-ARS-WRPIS; *Vogel et al. (2006)* | Iraq |
| BdTR10C | *B. distachyon* | Y | N | W6 39406 | USDA-ARS-WRPIS | Turkey |
| BdTR11I | *B. distachyon* | Y | N | W6 39426 | USDA-ARS-WRPIS; *Filiz et al. (2009)* | Turkey |
| BdTR13a | *B. distachyon* | Y | N | W6 39430 | USDA-ARS-WRPIS; *Filiz et al. (2009)* | Turkey |
| BdTR13C | *B. distachyon* | Y | N | W6 39432 | USDA-ARS-WRPIS; *Filiz et al. (2009)* | Turkey |
| BdTR1I | *B. distachyon* | Y | N | W6 39308 | USDA-ARS-WRPIS; *Filiz et al. (2009)* | Turkey |
| BdTR2B | *B. distachyon* | Y | N | W6 39314 | USDA-ARS-WRPIS; *Filiz et al. (2009)* | Turkey |
| BdTR2G | *B. distachyon* | Y | N | W6 39319 | USDA-ARS-WRPIS; *Filiz et al. (2009)* | Turkey |
| BdTR3C | *B. distachyon* | Y | N | W6 39332 | USDA-ARS-WRPIS; *Filiz et al. (2009)* | Turkey |
| BdTR5I | *B. distachyon* | Y | N | W6 39366 | USDA-ARS-WRPIS; *Filiz et al. (2009)* | Turkey |
| Foz1 | *B. distachyon* | Y | N | - | *Mur et al. (2011)* | Spain |
| Gaz8 | *B. distachyon* | Y | N | W6 39269 | USDA-ARS-WRPIS; *Vogel et al. (2009)* | Turkey |
| Kah1 | *B. distachyon* | Y | N | W6 39278 | USDA-ARS-WRPIS; *Vogel et al. (2009)* | Turkey |

*Table 1 continued on next page*

*Table 1 continued*

| Name | Species | Sequenced | Susceptible to *U. bromivora* | Accession number | Source | Country of origin |
|------|---------|-----------|------------------------------|------------------|--------|-------------------|
| Koz1 | *B. distachyon* | Y | N | W6 39284 | USDA-ARS-WRPIS; *Vogel et al. (2009)* | Turkey |
| Mur1 | *B. distachyon* | Y | N | - | *Mur et al. (2011)* | Spain |
| S8iiC | *B. distachyon* | Y | N | - | Ana Caicedo Lab, University of Massachusetts | Spain |
| ABR114 | *B. stacei* | Y | Y (2x)* | - | unknown | Spain |
| ABR113 | *B. hybridum* | Y | N | - | unknown | Portugal |
| ABR117, Bd117, Bd6 | *B. hybridum* | N | Y (2x)* | PI - 219965 | GRIN Germplasm | Afghanistan |
| Bal-P7 | *B. hybridum* | N | Y (2x)* | W6 - 39259 | GRIN Germplasm | Turkey |
| Bd23 | *B. hybridum* | N | Y (2x)* | PI - 287783 | GRIN Germplasm | Spain |
| Bd26 | *B. hybridum* | N | Y (2x)* | PI - 372187 | GRIN Germplasm | Uruguay |
| Bd28 | *B. hybridum* | N | Y (many)* | PI - 533015 | GRIN Germplasm | Australia |
| Bd4 | *B. hybridum* | N | Y (2x)* | PI - 208216 | GRIN Germplasm | South Africa |
| Bd8 | *B. hybridum* | N | Y (2x)* | PI - 219971 | GRIN Germplasm | Afghanistan |
| Isk-P4 | *B. hybridum* | N | Y (2x)* | W6 - 39273 | GRIN Germplasm | Turkey |

Y = yes, N = no; * = times tested

necessary result of the mating type bias and ensures an efficient re-entry into the pathogenic stage of the life cycle. At the same time, the haplo-lethal allele that is likely linked to the MAT-2 locus promotes an inbreeding mode of *U. bromivora* with direct consequences on speciation, genome size, and selection on recessive alleles as it has been observed in other organisms (*Wright et al., 2008*; *Ellegren and Galtier, 2016*; *Joly, 2011*).

## Isolation of a haplo-viable *U. bromivora* MAT-2 strain

Although the observed mating type bias is an interesting biological phenomenon, it is an undermining factor for the use of *U. bromivora* as genetically accessible model system to study biotrophic interactions. It creates a situation where one mating partner can only be maintained in the pathogenic, dikaryotic stage together with its compatible mating type and, as a consequence, it can neither be cultured axenically nor transformed under laboratory conditions. Depending on the physical distance of the MAT-2 locus to the haplo-lethal allele that causes the mating type bias, homologous recombination can lead to an uncoupling and the formation of a haplo-viable MAT-2 strain. Therefore, we screened for such viable haploid MAT-2 recombinants by using a pooled infection assay where a pool of spore-derived progeny was grown saprophytically and used for re-infection of *Brachypodium* sp. (for details see Material and Methods). By pursuing this approach, we identified one MAT-2 strain that we named UB2. This strain, which is not derived from the same spore as UB1, retained the capability to mate with the *U. bromivora* MAT-1 strain UB1, which we isolated in a classical spore germination assay. Moreover, upon inoculation of *Brachypodium* sp., UB2 along with UB1 formed viable spores demonstrating a successful completion of their life cycle. Infection symptoms of UB1xUB2-inoculated plants were indistinguishable from infected spikelets of spore-inoculated plants (data not shown).

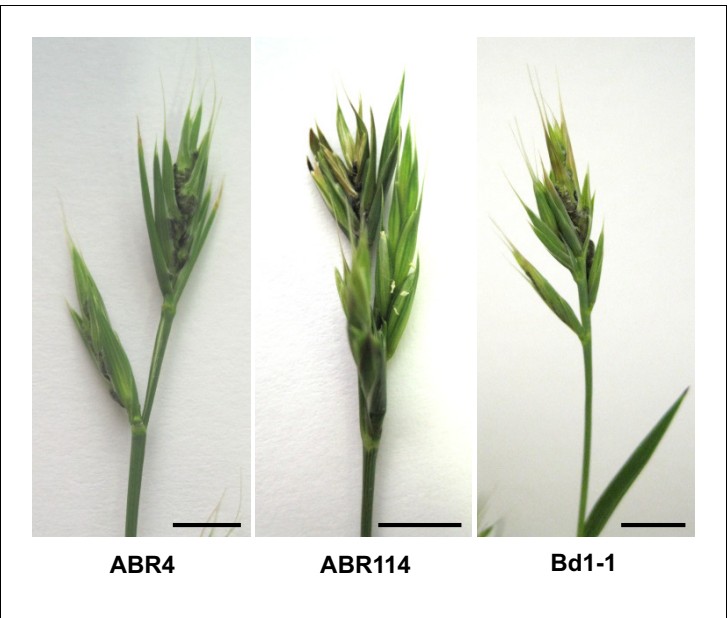

**Figure 4.** ABR4, ABR114, and Bd1-1 are susceptible to *U. bromivora*. Representative pictures of infected ABR4, ABR114, and Bd1–1 spikelets. Sequenced *B. distachyon* accessions ABR4 and Bd1-1 as well as the sequenced *B. stacei* accession ABR114 were inoculated with *U. bromivora* spore material and screened for infection symptoms. Upon flowering macroscopic infection symptoms could be observed as spore-filled spikelets. This figure relates to *Table 1*. Scale bars: 1 cm.

## Identification of a sequenced compatible host of *U. bromivora*

Phylogenetic analysis recently led to a taxonomy split within the *Brachypodium* lineages, separating *B. distachyon* with cytotype 2n = 10 from *Brachypodium stacei* (2n = 20) and the allotetraploid *Brachypodium hybridum* (2n = 30) (*Catalan et al., 2012*). The spontaneous infection event by *U. bromivora* reported by *Barbieri et al. (2011)* occurred in the *B. hybridum* accession Bd28. The *B. hybridum* accession Bd28 undergoes self-fertilization, grows fast, and is easy to handle (*Barbieri et al., 2011*). One slightly complicating fact is its allotetraploid genetic background but its genome is currently being sequenced and will be available to the community in near future (J. Vogel, unpublished). In order to identify additional *Brachypodium* sp. accessions susceptible to *U. bromivora*, in total, 39 accessions were infected with spores and analyzed for macroscopic infection symptoms in the spikelets. Whereas *B. distachyon* Bd21, along with 27 other tested accessions, did not show infection symptoms, we found eleven accessions to be fully susceptible to *U. bromivora* (*Table 1*). The finding that susceptible host plant accessions originate from Europe, Asia, Africa, Australia as well as South America (*Table 1*) underlines that *U. bromivora* is considered as a cosmopolite (*Bauch, 1925*). The natural host range of *U. bromivora* comprises various species of the genera *Agropyron, Austrofestuca, Brachypodium* including *B. distachyon, Bromus, Critesion, Elymus, Festuca, Hordeum, Lolium, Sitanion* and *Trachynia* (*Bauch, 1925*; *Fisher and Holton, 1957*; *Vanky, 2011*) and our experiments can confirm at least three host species, *B. distachyon, B. hybridum* and *B. stacei*. Among the susceptible accessions are the *B. distachyon* accessions ABR4, originally collected from Southern Spain, and Bd1-1, an inbred line, as well as the *B. stacei* accession ABR114 (*Figure 4*). The genomes of all three susceptible diploid accessions have been sequenced (J. Vogel, unpublished), providing the foundation for valuable tools and studies of our novel plant pathosystem. While, to our knowledge, ABR4 has not been tested for susceptibility to other important fungal pathogens, Bd1-1 shows resistance to the wheat pathogen *Zymoseptoria tritici* as well as to *Puccinia graminis* (*Figueroa et al., 2013*; *O'Driscoll et al., 2015*). Therefore, Bd1-1 represents a suitable model accession to study compatible interactions with *U. bromivora* as well as incompatible interactions with *P. graminis* and *Z. tritici*. Moreover, the comparison of susceptible *B. distachyon* accessions such as Bd1-1 and ABR4, with the resistant accession Bd21 allows the study of

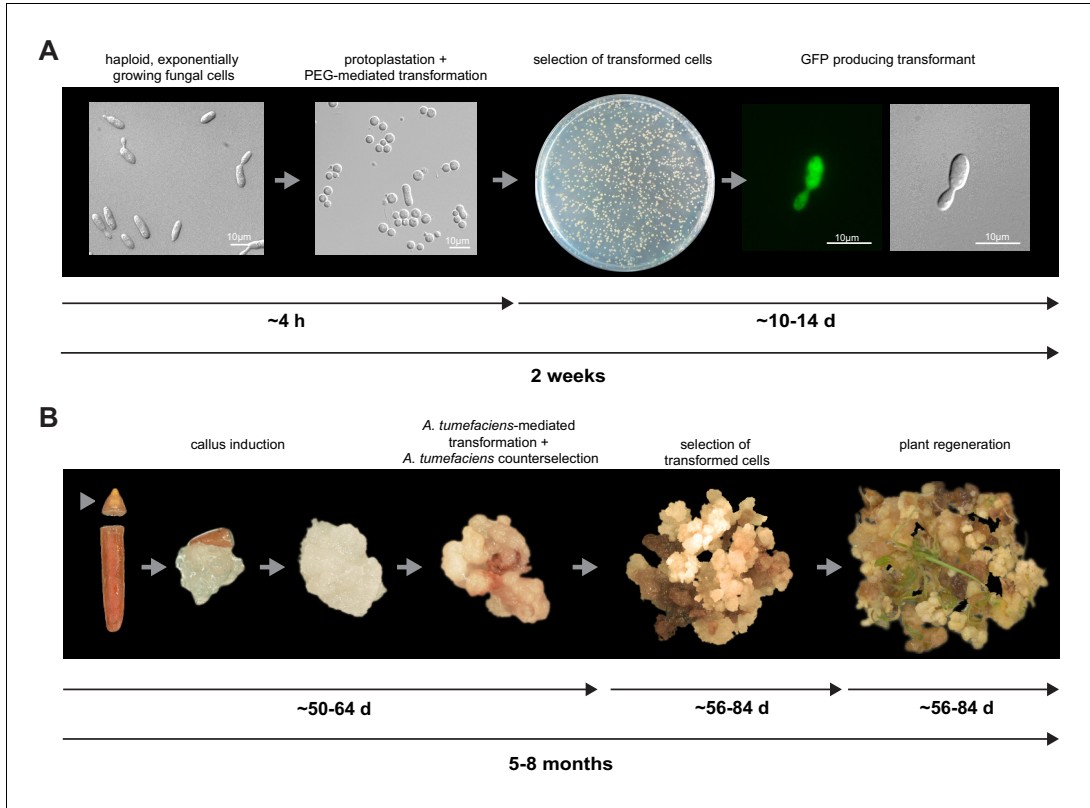

**Figure 5.** Establishment of transformation for the fungal pathogen *U. bromivora* and its host plant *B. hybridum* Bd28. (**A**) Schematic representation and timeline of protoplastation and transformation of *U. bromivora* UB1 with the autonomously replicating pNEBuC-GFP plasmid conferring Carboxin resistance and encoding GFP. (**B**) Schematic representation and timeline of *A. tumefaciens*-mediated transformation of *B. hybridum* Bd28. Illustrations are not to scale.

The following figure supplements are available for figure 5:

**Figure supplement 1.** Several resistance markers can be employed for selection of *U. bromivora* transformants.

**Figure supplement 2.** UB2 can mate with UB1-GFP, form filaments, and produce viable spores.

**Figure supplement 3.** Microscopy of transgenic Bd28 eCFP-SKL marker line.

compatible and incompatible interactions with *U. bromivora* without switching pathosystems. The discovery of several *Brachypodium* accessions resistant to *U. bromivora*, comprising accessions of *B. distachyon*, *B. stacei*, and *B. hybridum*, suggests that resistance likely evolved in a common ancestor of *B. distachyon* and *B. stacei*. As there is evidence that *B. hybridum* is the product of an interspecies cross between these two diploid taxa (*Petersen et al., 2011*), this scenario would suggest that smut resistance was independently lost several times in each taxon. Alternatively, resistance based on single or even multiple factors could have evolved independently several times in the different closely related species. Within the limits of a relatively small sample size we observe a trend of *B. hybridum* accessions to be susceptible to *U. bromivora*. This could be due to suppression of resistance in polyploid genomes, a previously described phenomenon found in tetra- and hexaploid wheat (*Knott, 2000*; *Kerber, 1991*).

Since Bd28 is an excellent host for *U. bromivora* and needs only very short vernalization to induce flowering and to show infection symptoms, we used this accession for the establishment of a model host. To this end, we developed an efficient growth and infection method providing high germination rate, reliable floral induction, and high infection rate by *U. bromivora* (for details see Material and Methods). These protocols could be also applied for the diploid and sequenced accession

ABR4. However, in contrast to Bd28, ABR4 requires a minimum of four weeks vernalization to induce flowering and needs six weeks of vernalization to induce fast bulk flowering. 95 days after the initiation of vernalization, flowering of ABR4 is completed and we observed infection rates of almost 100%. Due to its shorter vernalization requirement, Bd28 can complete its life cycle in approximately 70 days. Interestingly, different time points of spore inoculation as well as the spore load seem to lead to varying infection efficiency. We regularly observed infected plants showing a few healthy spikelets beside spore-filled ones (data not shown). The presence of healthy spikelets might be the evolutionary result of a sustainable virulence strategy of this host-specific, biotrophic pathogen to avoid complete sterilization of its host and therefore its local extinction.

## Transformation establishment for *U. bromivora* and its host *B. hybridum* Bd28

The establishment of transformation protocols for both the pathogen and its host plant is an important requirement to employ *U. bromivora* and *Brachypodium* sp. as a genetically accessible biotrophic model system.

For initial transformation tests of *U. bromivora*, we used the self-replicating pNEBuC-GFP and pNEBuC-mCherry-HA vectors, which were developed for *U. maydis* (*Brachmann, 2001*). These plasmids contain a gene conferring Carboxin resistance and a gene encoding either the green fluorescent protein (GFP-HA) or mCherry-HA protein. *GFP-HA* or *mCherry-HA* are under control of the artificial *otef* promoter, which is constitutively active in axenic culture (*Spellig et al., 1996*). Expression of these genes allows a fast readout of the transformation efficiency. Although different transformation protocols were tested, PEG-mediated protoplast transformation turned out to be most efficient (average transformation efficiency: 314 colonies per µg plasmid DNA; *Figure 5A*). Since appropriate selection markers are essential for high-efficiency transformations, we concomitantly tested three different antibiotics and selection markers. While wild type cells could not be propagated, transformants harboring self-replicating plasmids conferring Carboxin, Geneticin G418, or Hygromycin resistance were able to grow on respective selection media (*Figure 5—figure supplement 1*). We also tested integrative constructs for various loci such as the mating type region or the predicted *pep1* gene locus, encoding an effector ortholog that has been shown to contribute to virulence in *U. maydis* (*Doehlemann et al., 2009*). However, our results suggest, that in contrast to *U. maydis*, DNA uptake or other steps during transformation seem to be less efficient in *U. bromivora* preventing high transformation rates and strongly reduced the number of stable integration events. Therefore, we conducted a restriction enzyme mediated integration (REMI) approach which should promote genomic integrations (*Bölker et al., 1995*). By employing this technique, we obtained a UB1 derivative with a stable integration of *GFP* driven by the *otef* promoter (UB1-GFP). To ensure that after REMI mutagenesis, UB1-GFP has retained its capability to infect, we inoculated germinating Bd28 caryopses with a mixture of UB1-GFP and UB2. Spore-filled spikelets and GFP producing progeny derived from these spores demonstrated a successful infection and completion of the fungal life cycle (*Figure 5—figure supplement 2*).

To establish transformation for the *B. hybridum* accession Bd28, we adapted a Bd21 transformation protocol (*Vain et al., 2008*) for its specific needs (*Figure 5B*). As a proof of principle, we generated transgenic lines harboring a fluorescently tagged peroxisome-targeting construct (eCFP-SKL). On the one hand, this construct provides a fast visual read out of a successful transformation; on the other hand, respective transgenic lines might be a valuable tool for studying cell biological questions in *Brachypodium* sp. (*Figure 5—figure supplement 3*).

## The genome of *U. bromivora*

By comparative rDNA analysis, *U. hordei* has been shown to be the most closely related smut to *U. bromivora* that has been sequenced to date (*Stoll et al., 2005*). The presence of extensive repetitive elements and transposable elements (TE) in the *U. hordei* genome complicated its genome assembly (*Laurie et al., 2012*). To circumvent possible assembly problems with the related *U. bromivora* genome, we performed Single Molecule Real-Time (Pacific Biosciences, Menlo Park, CA) sequencing of UB1, the isolated MAT-1 strain. PacBio sequencing has been shown to deliver long reads, facilitating fast and accurate genome assembly. After subread filtering, 376,645 reads (3.1 Gb total) with 84.5% accuracy and a mean length of 8,186 bp (approximately 154x coverage of the ~20.7 Mb

**Table 2.** Genome comparison of sequenced smut fungi.

| | U. bromivora | U. maydis[1] | S. reilianum[2] | S. scitamineum[3] | U. hordei[4] | M. pennsylvanicum[5] |
|---|---|---|---|---|---|---|
| **Assembly statistics** | | | | | | |
| Total contig length (Mb) | | 19.7 | 18.2 | 19.5 | 20.6 | 19.2 |
| Total scaffold length (Mb) | 20.5 | 19.8 | 18.4 | 19.6 | 21.15 | 19.2 |
| Average base coverage | 154x | 10x | 20x | 30x | 25x | 339x |
| $N_{50}$ contig (kb) | | 127.4 | 50.3 | 37.6 | 48.7 | 43.4 |
| $N_{50}$ scaffold (kb) | 877 | 817.8 | 738.5 | 759.2 | 307.7 | 121.7 |
| Chromosomes | 23 | 23 | 23 | | 23 | |
| GC-content (%) | 52.4 | 54 | 59.7 | 54.4 | 52 | 50.9 |
| coding (%) | 54.4 | 56.3 | 62.6 | 57.8 | 54.3 | 54 |
| non-coding (%) | 49.4 | 50.5 | 54.3 | 51.1 | 43.4 | 46.9 |
| **Coding sequence** | | | | | | |
| Percent coding (%) | 59.8 | 61.1 | 65.9 | 62 | 57.5 | 56.6 |
| Average gene size (bp) | 1699 | 1836 | 1858 | 1819 | 1705 | 1734 |
| Average gene density (gene/kb) | 0.35 | 0.34 | 0.36 | 0.34 | 0.33 | 0.33 |
| Protein-coding genes | 7233 | 6786 | 6648 | 6693 | 7113 | 6279 |
| Exons | 11154 | 9783 | 9776 | 10214 | 10907 | 9278 |
| Average exon size | 1101 | 1230 | 1221 | 1191 | 1107 | |
| Exons/gene | 1.5 | 1.44 | 1.47 | 1.5 | 1.53 | 1.48 |
| tRNA genes | 133 | 111 | 96 | 116 | 110 | 126 |
| **Secretome** | | | | | | |
| Predicted secreted proteins | 409 | 485 | 461 | 466 | 405 | 300 |
| **Non-coding sequence** | | | | | | |
| Introns | 3921 | 2997 | 3103 | 3521 | 3161 | 2999 |
| Introns/gene | 0.54 | 0.44 | 0.46 | 0.53 | 0.44 | 0.48 |
| Average intron length (base) | 163 | 142 | 144 | 130.1 | 141 | 191.4 |
| Average intergenic distance (bp) | 1054 | 1127 | 929 | 1114 | 1186 | 1328 |
| **Repeat sequences** | | | | | | |
| DNA Transposon | 1.89% | 0.29% | 0.13% | 0.25% | 0.89% | 0.29% |
| LINE | 4.38% | 0.35% | 0.04% | 0.27% | 4.62% | 0.40% |
| SINE | 0.18% | 0.05% | 0.03% | 0.05% | 0.27% | 0.10% |
| LTR Retrotransposon | 5.83% | 1.15% | 0.13% | 0.69% | 4.82% | 1.17% |
| Unclassified non LTR-Retrotransposon | 0.06% | 0.02% | 0.01% | 0.01% | 0.10% | 0.032 |
| Unclassified Retrotransposon | 2.03% | 0.21% | 0.12% | 0.29% | 1.47% | 0.39% |
| Unclassfied | 0.06% | 0.08% | 0.02% | 0.08% | 0.38% | 0.04% |
| Total TE class | 14.33% | 2.11% | 0.45% | 1.60% | 11.84% | 2.32% |
| Simple sequence repeats | 1.31% | 1.75% | 2.00% | 1.59% | 1.59% | 1.54% |
| Total excl. Tandem repeats | 15.72% | 3.90% | 2.49% | 3.23% | 13.56% | 3.95% |
| Tandem repeats | 5.14% | 4.22% | 6.97% | 4.54% | 5.20% | 5.16% |
| Total repeat coverage | 18.51% | 6.70% | 8.26% | 6.68% | 16.45% | 6.72% |

[1]*Kämper et al., 2006*; [2]*Schirawski et al., 2010*; [3]*Dutheil et al., 2016*; [4]*Laurie et al., 2012*; [5]*Sharma et al., 2014*

genome) were obtained. Assembly and polishing resulted in 25 contigs of which 23 showed good synteny with the optically mapped synthetic chromosomes of *U. hordei* (*Laurie et al., 2012*). Based on this, *U. bromivora* has 23 chromosomes (20.5 Mb), the mitochondrial genome and an unassigned

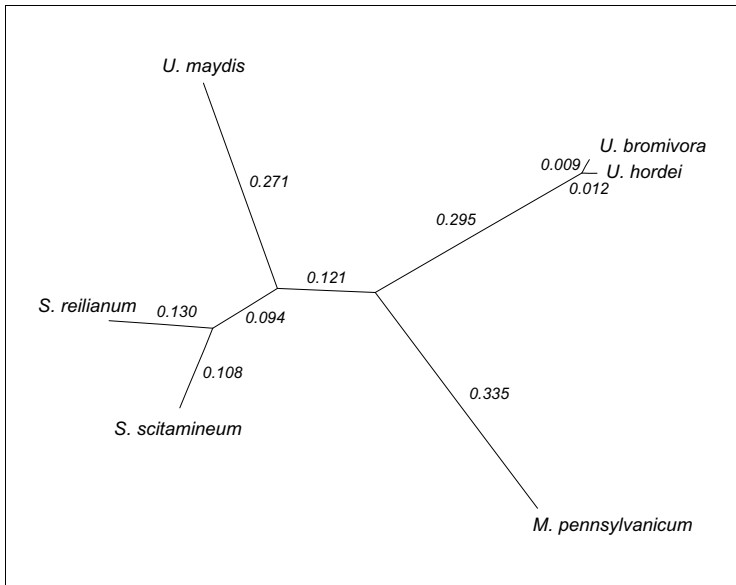

**Figure 6.** Phylogeny of *U. bromivora* and related smuts. Unrooted phylogeny created from 4,947 one-to-one orthologs. Branch lengths represent the mean number of substitutions per DNA site. Terminal branch lengths for *U. bromivora* and *U. hordei* are not to scale.

smaller contig. Gene model prediction led to the identification of 7,233 protein coding genes (*Table 2*). All gene models were manually curated by extensive comparative analysis to the existing Ustilaginaceae annotations and 82.3% of all gene models were confirmed by assembled transcripts obtained from Illumina-based RNA-seq of axenically grown UB1. With 7,233 protein coding genes, *U. bromivora* harbors 120 more genes than the close relative *U. hordei* that has an assembled genome 0.65 Mb larger than *U. bromivora* (*Laurie et al., 2012*). The compactness of the *U. bromi-vora* genome is underscored by the fact that 67.9% of the genes are intron-less, 19.3% have one intron, whereas only 12.8% are predicted to contain multiple introns.

To assess the completeness of the *U. bromivora* genome, a BLAST search was performed with highly conserved core genes present in higher eukaryotes (*Aguileta et al., 2008*; *Parra et al., 2009*). From the expected 248 single-copy orthologs extracted from 21 genomes (*Parra et al., 2009*), 247 are present in the *U. bromivora* genome (missing KOG1468, translation initiation factor eIF-2B), indicating that >99% of the gene space is covered by the assembly.

To assess the difference of UB1 and its compatible mating partner UB2 on a genomic level, we performed Illumina 125 paired-end sequencing of UB2 and single nucleotide polymorphism (SNP) calling. By using stringent parameters, we identified 1,323 SNPs between the two strains (*Figure 7— source data 1*). 783 of them are found in coding sequences and 429 lead to non-synonymous muta-tions. Since UB1 and UB2 were independently isolated from different spores, SNPs might be a result of the prevalent inbreeding due to intratetrad mating which over time leads to significant differences between progenies of different spore tetrads.

We constructed a phylogenetic tree using one-to-one orthologous genes as identified by OrthoMCL (*Figure 6*). The resulting phylogeny shows *Sporisorium reilianum* and *Sporisorium scitami-neum* forming one cluster and *U. bromivora* and *U. hordei* forming a second cluster. *U. maydis* shows a closer relationship to the tested species of the *Sporisorium* genus than to the ones of the *Ustilago* genus that are more closely related to *Melanopsichium pennsylvanicum*. This is in agree-ment with the relationships found by *Stoll et al. (2005)* and *Sharma et al. (2014)* . The phylogeny also illustrates the close relationship between *U. hordei* and *U. bromivora*, which share orthologs for a large number of genes and between which protein sequence conservation is much higher than between other fungi (data not shown).

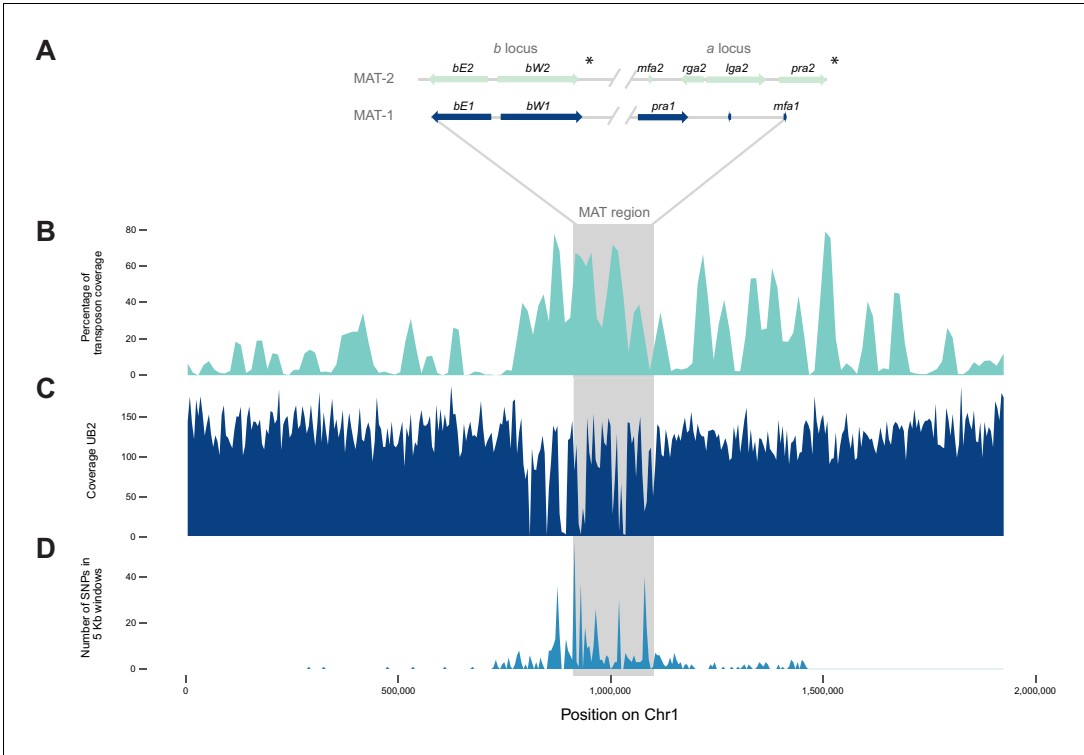

**Figure 7.** The mating type chromosome (chromosome 1) of *U. bromivora* UB1 (MAT-1) and UB2 (MAT-2). (A) Schematic representation of the *a* locus (encoding the predicted pheromone receptor system) and the *b* locus (encoding the putative heterodimeric transcription factor for pathogenic development) of UB1 (MAT-1 strain) and UB2 (MAT-2 strain) according to de novo assembly of both strains. Potential *rga2* and *lga2* orthologs, encoding proteins for uniparental mitochondrial inheritance (*Fedler et al., 2009*), are located in the *a2* region between *mfa2* and *pra2*. *Due to rearrangements in the mating type region of UB2, the orientation of *a2* and *b2* locus could not be exactly determined. However, data suggest an inversion of the *a2* locus. (B) The mating type region of UB1 is enriched for transposable elements. Graph depicts percentage of transposon coverage in 25 kb windows occurring every 12.5 kb along the chromosome. (C) Mapping of UB2 to the UB1 reference genome shows large non-mapped stretches in the MAT-1 locus indicating sequence differences in this region between MAT-1 and MAT-2. (D) Enrichment of single nucleotide polymorphisms (SNPs) in and around the mating type region between the genomes of UB1 and UB2. Number of SNPs is shown in 5 kb windows.

The following source data and figure supplements are available for figure 7:

**Source data 1.** List of Single Nucleotide Polymorphisms (SNPs) identified in UB2.

**Figure supplement 1.** Transposon content along *U. bromivora* chromosomes.

**Figure supplement 2.** Genes of the mating type regions are up to 98% identical between *U. bromivora* and *U. hordei*.

## The mating type chromosome

The mating type locus of *U. bromivora* UB1 is located on chromosome 1. We identified genes encoding a putative pheromone receptor (UBRO_03901) and pheromone (UBRO_03899) as well as bEast (UBRO_00885) and bWest (UBRO_00887) orthologs encoding a putative homeodomain-transcription factor (*Figure 7A*). The pheromone/receptor genes (*a* locus) are separated from the *bEast* and *bWest* genes (*b* locus) by a 183 kb long region highly enriched in transposable elements (TE; 39.85% compared to an average of 17.18% over the length of the chromosome and to an overall genome average of 14.33%; *Figure 7B*, *Figure 7—figure supplement 1*). This bipolar mating system resembles the structural organization found in the close relative *U. hordei* although the region

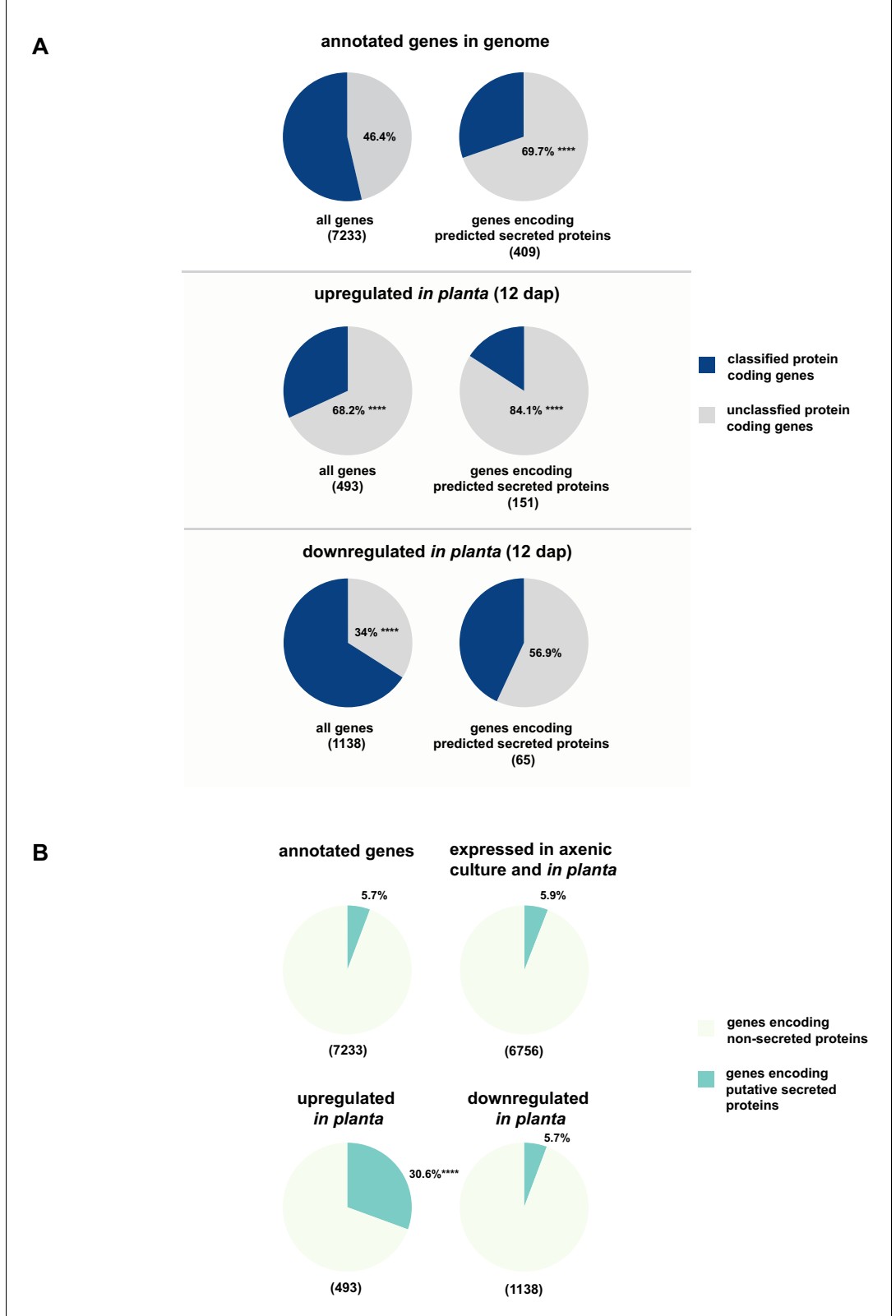

**Figure 8.** Enrichment of classes of interest among predicted secreted proteins and *in planta* differentially expressed transcripts. (**A**) Proportions of genes encoding unclassified proteins based on FunCat classification within all annotated *U. bromivora* genes, within genes encoding secreted proteins, within all genes up- and downregulated *in planta* twelve days after planting (dap) as well as within the genes that are differentially regulated and encode secreted proteins. (**B**) Proportions of genes encoding predicted secreted proteins within all annotated *U. bromivora* genes, within all genes

*Figure 8 continued on next page*

*Figure 8 continued*

found to be expressed in axenic culture and *in planta*, and within genes significantly up- or downregulated *in planta*. Fisher exact test was used to test whether the proportion of selected genes within a given class differs significantly from the proportion within all annotated genes; ****p-value < 0.0001, ***p-value < 0.001. The total number of genes is shown in brackets below each chart.

The following figure supplement is available for figure 8:

**Figure supplement 1.** Functional categorization of putatively secreted proteins and all proteins encoded in the genome.

between the *a* and *b* locus in *U. hordei* is, at ~500 kb, larger (*Figure 7—figure supplement 2*) (*Bakkeren and Kronstad, 1994*).

Mapping of Illumina reads from UB2 to the UB1 reference genome and subsequent SNP calling showed that the mating type region flanked by the *a* and *b* loci is highly diverse between the two compatible strains, as evidenced by segments depleted of reads, rearrangements and a high number of SNPs in the regions covered (*Figure 7C,D*; *Figure 7—source data 1*). We calculated for the mating type region a SNP frequency of 1.8E-03 SNPs $bp^{-1}$ compared to a frequency of 6.5E-05 SNPs $bp^{-1}$ for the entire genome. This sequence divergence is likely a result of recombination suppression in the mating type region leading to a bipolar mating system at the cost of losing a recombination-based repair in this region.

## Analysis of genes encoding putatively secreted proteins

One class of proteins which is of special interest for biotrophic interactions is secreted proteins. In addition to functions such as cell wall remodeling and hydrolytic enzymes for substrate degradation, this class includes effector proteins, which might directly impact the outcome of the biotrophic interaction with the host and often target the host defense system or its metabolism (*Asai and Shirasu, 2015*). Several criteria had to be fulfilled for a predicted *U. bromivora* protein to be considered a

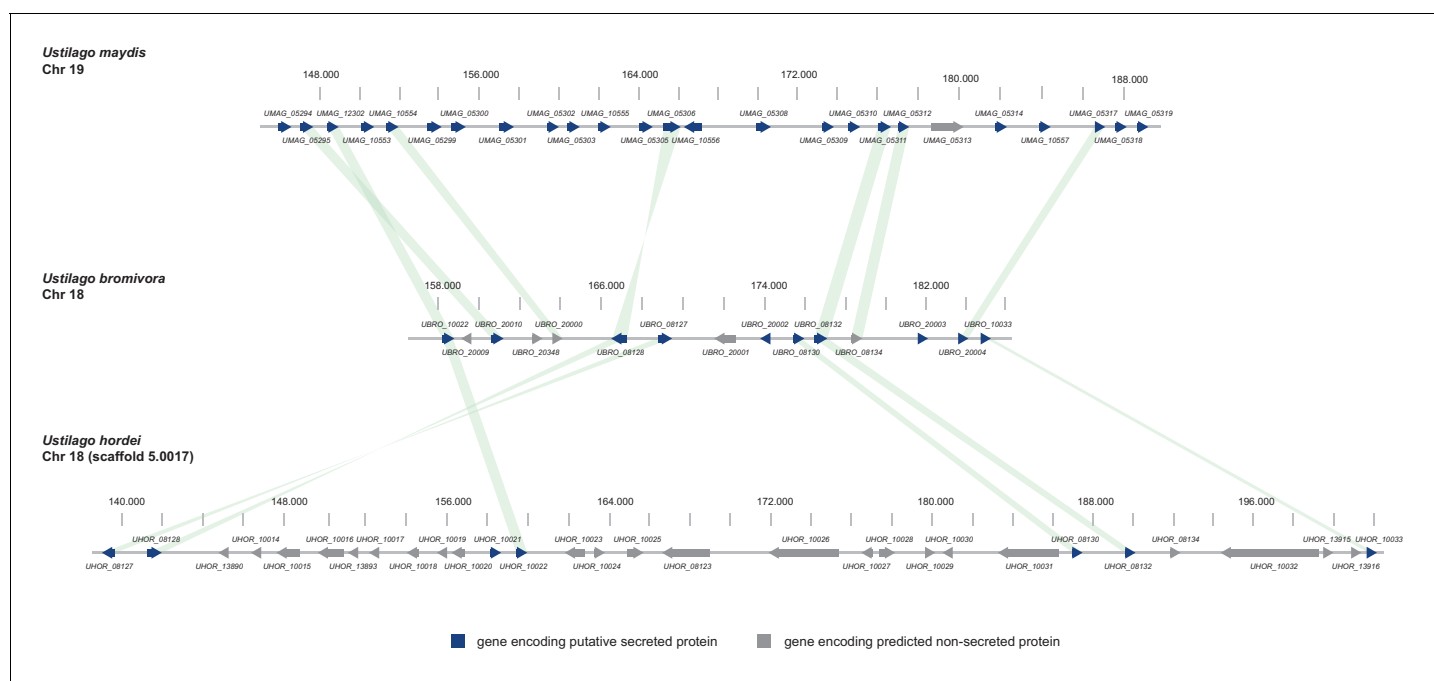

**Figure 9.** Gene by gene comparison between the largest secreted virulence cluster of *U. maydis* (cluster 19) with the corresponding region in the *U. bromivora* and *U. hordei* genome on chromosome 18. The scheme depicts *U. maydis* cluster 19 on chromosome 19, the predicted *U. bromivora* cluster 18 on chromosome 18 and the corresponding region on chromosome 18 (scaffold 5.0017) of *U. hordei*. Syntenic orthologs between *U. maydis* and *U. bromivora* as well as between *U. bromivora* and *U. hordei* are connected with a green bar. Genes encoding predicted non-secreted proteins are displayed in grey, genes encoding putative secreted proteins in blue.

secreted protein: it had to contain a signal peptide, fewer than two transmembrane helices, no endoplasmatic reticulum (ER) retention signal, and it had to be predicted not to target mitochondria. These strict criteria led to the prediction of 409 secreted proteins encoded in the *U. bromivora* genome.

## The majority of putatively secreted proteins are of unknown function and their transcripts are overrepresented *in planta*

While for 46.3% of all *U. bromivora* proteins we could not assign a potential function based on BLAST sequence similarity ('unclassified proteins'), this fraction increases to 69.7% for the proteins that are predicted to be secreted. This makes proteins of unknown function the most significantly enriched functional category of *U. bromivora* secreted proteins (*Figure 8A*, *Figure 8—figure supplement 1*).

To provide insights into expression patterns of the genes encoding putatively secreted proteins in *U. bromivora* during saprophytic growth and *in planta*, we conducted RNA-seq analyses. To this end, we isolated RNA from axenic UB1 culture and from stems of twelve day old *B. hybridum* Bd28 plants that were spore-inoculated with *U. bromivora*. Among the 6,756 transcripts found to be expressed in our dataset, 493 were significantly upregulated *in planta* compared to axenic culture (logFC > 2, adjusted p-value < 0.1), while 1,138 transcripts were significantly downregulated. Notably, transcripts predicted to encode secreted proteins are significantly enriched among the upregulated transcripts compared to all annotated genes (30.8% compared to 5.7%; Fisher exact test, p-value ≤ 2.2E-16; *Figure 8B*). Moreover, we could show by functional protein classification that among the putatively secreted proteins that are induced *in planta*, unclassified proteins are significantly overrepresented (84.1% compared to 46.4% within all annotated genes; p-value = 2.2E-16; *Figure 8A*). In contrast, in the subset of down-regulated genes, a functional overrepresentation of genes encoding secreted, unclassified proteins could not be observed. Our findings are in line with the observation that the vast majority of effector proteins are so far functionally not characterized. At the sequence level these proteins are only poorly conserved between distant fungal pathogens making the prediction of protein function difficult.

## The majority of putative *U. bromivora* effectors are not clustered

In the case of *U. maydis,* many of the small secreted protein encoding genes are organized in pathogenicity clusters and a large number of them were shown to play a role in virulence (*Kämper et al., 2006*). To assess the presence of potential pathogenicity clusters in the *U. bromivora* genome, we defined clusters as containing at least three adjacent genes encoding predicted secreted proteins or three genes encoding secreted proteins interrupted by maximal one single non-secreted protein-coding gene. These in comparison to the analysis of *Kämper et al. (2006)* relatively relaxed criteria led to the identification of only eleven secretion clusters comprising a total of 10.51% of all putative secreted proteins of *U. bromivora*. Although some of the putative effector clusters identified in *U. maydis* also show a syntenic organization in the *U. bromivora* genome, many others do not and the vast majority of the predicted secreted proteins of *U. bromivora* are not clustered. Among all clusters, the largest is cluster 18, related to *U. maydis* cluster 19. Whereas cluster 19 (as located on chromosome 19) in *U. maydis* comprises 24 putative effector genes (*Brefort et al., 2014*), the syntenic cluster in *U. bromivora* comprises less than half the number of putative secreted protein encoding genes and is located on chromosome 18 (*Figure 9*). In the close relative *U. hordei* the clustering of genes encoding predicted secreted proteins in this region is even less compact and disrupted to a greater extent than in *U. bromivora* (*Figure 9*) (*Laurie et al., 2012*). Our analysis of the *U. bromivora* clusters further showed that the syntenic genes missing from cluster 18 have not moved to other parts of the genome but are simply absent from *U. bromivora* and occur only in *U. maydis* or are shared between *U. maydis*, *S. reilianum*, and *S. scitamineum* (data not shown).

## Orthologous and orphan genes

One indication of how suitable a specific model system is for generalizing knowledge to other related species is the presence of orthologous genes. Excluding mitochondrial genes, among the 7,193 genes present in the *U. bromivora* genome, 5,121 one-to-one orthologs were predicted with *U. maydis*, *U. hordei*, *S. scitamineum*, *S. reilianum,* and *M. pennsylvanicum*. These one-to-one

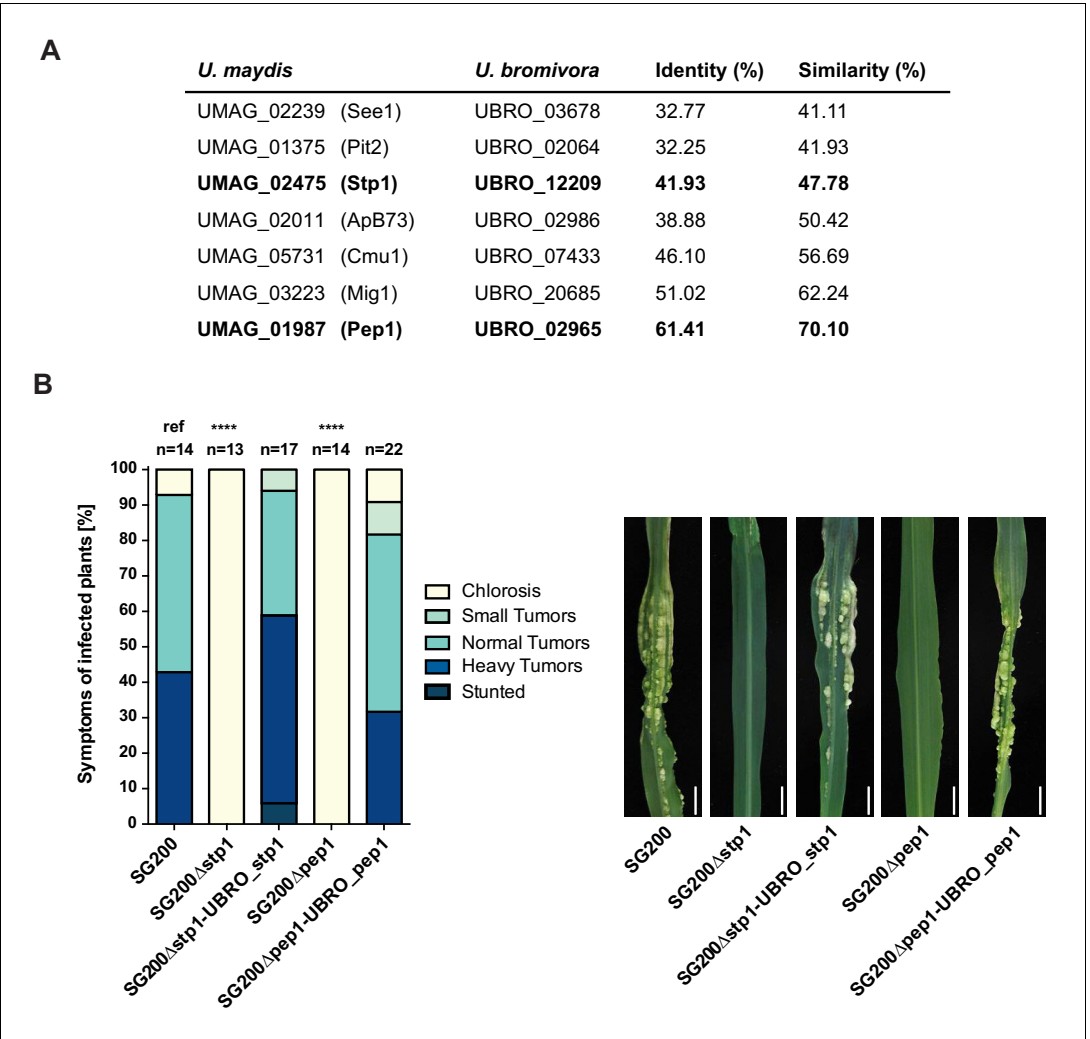

**A**

| U. maydis | | U. bromivora | Identity (%) | Similarity (%) |
|---|---|---|---|---|
| UMAG_02239 | (See1) | UBRO_03678 | 32.77 | 41.11 |
| UMAG_01375 | (Pit2) | UBRO_02064 | 32.25 | 41.93 |
| **UMAG_02475** | **(Stp1)** | **UBRO_12209** | **41.93** | **47.78** |
| UMAG_02011 | (ApB73) | UBRO_02986 | 38.88 | 50.42 |
| UMAG_05731 | (Cmu1) | UBRO_07433 | 46.10 | 56.69 |
| UMAG_03223 | (Mig1) | UBRO_20685 | 51.02 | 62.24 |
| **UMAG_01987** | **(Pep1)** | **UBRO_02965** | **61.41** | **70.10** |

**Figure 10.** Testing orthologs of core effectors for functional interchangeability between *U. bromivora* and *U. maydis*. (A) List of known effector orthologs in *U. bromivora* and *U. maydis* and their amino acid identity and similarity. Identity shows the percentage of identical positions in the alignment, taking gaps into account. Percentage identity = 100 (identical positions / length of alignment). Similarity gives a measure of how similar two protein sequences are to one another based on the physical and chemical properties of their amino acids. Sequences were aligned using T-Coffee and identity and similarity scores were given by SIAS (Sequence Identity and Similarity; http://imed.med.ucm.es/Tools/sias.html). (B) Core-effector mutants of *U. maydis* (SG200Δstp1 and SG200Δpep1) can be complemented with the respective *U. bromivora* ortholog. Disease symptoms of infected plants were scored at twelve days post inoculation (dpi) according to *Kämper et al., 2006*. The darker the color, the more severe the symptoms. Numbers of infected plants are indicated above each column. p-values are calculated by Fisher exact test, MTC by Benjamini-Hochberg algorithm, ****p<0.0001. Leaves of representative plants twelve days after inoculation with indicated strains are shown next to the stacked bar plot. Scale bars: 1 cm.

orthologs are genes which have one ortholog in each of the other species and no paralogs. *M. pennsylvanicum* and *U. bromivora* share 5,470 orthologs. In contrast, *U. bromivora* shares 5,841 orthologs with *U. maydis* and 6,180 orthologs with *U. hordei*. The lower number of orthologs in *M. pennsylvanicum* is in line with the observation that this smut genome has lost genes that might be associated with a switch from a monocot to a dicot host plant (*Sharma et al., 2014*).

Among the 409 *U. bromivora* predicted secreted proteins, we found 216 of them among the 5,121 one-to-one orthologs. All of the *U. maydis* effectors functionally characterized or described in the literature, including Tin2, See1, Cmu1, Pep1, Pit2 and Mig1 (*Djamei et al., 2011*; *Redkar et al., 2015a*, *2015b*; *Hemetsberger et al., 2012*; *Doehlemann et al., 2009*; *Mueller et al., 2013*;

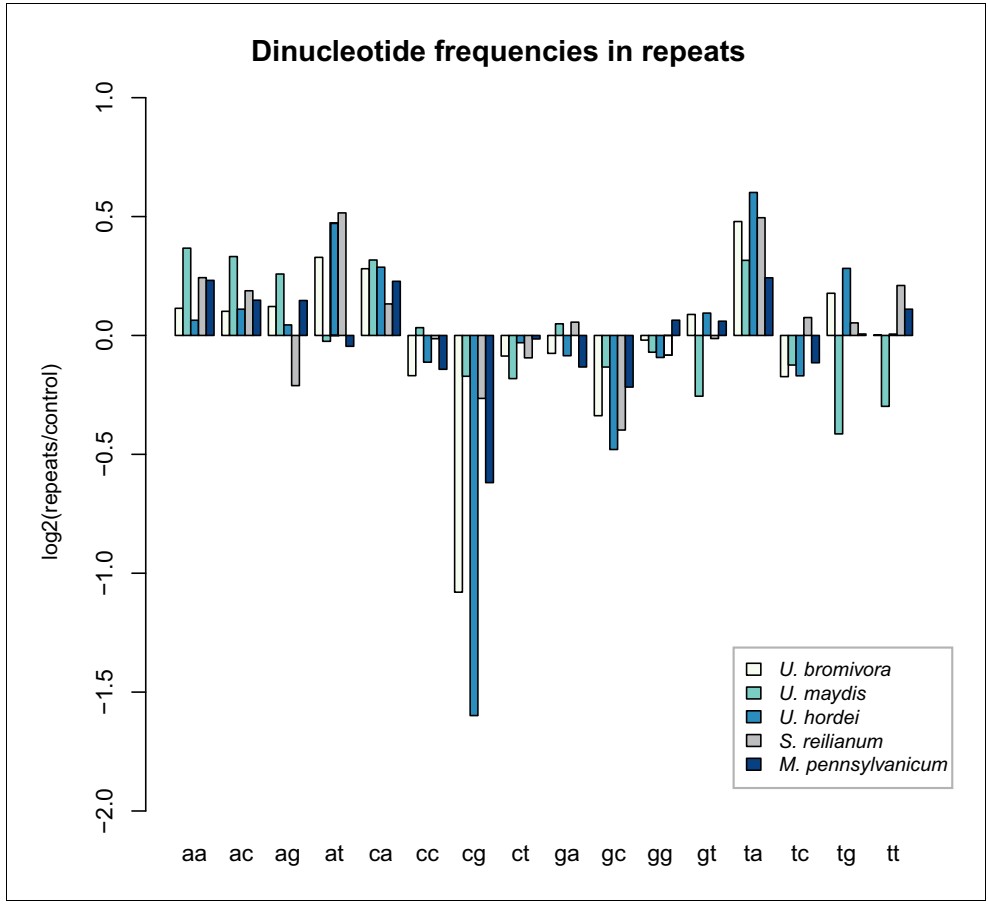

**Figure 11.** *U. bromivora* shows a decrease in the occurrence of CpG dinucleotides. Dinucleotide frequencies in repeat regions were determined using RIPCAL and were compared to those of control regions. A noticeable decrease in the occurrence of CpG dinucleotides was detectable for both *U. hordei* and *U. bromivora* as well as, to a lesser extent, *M. pennsylvanicum*.

*Basse et al., 2000*; *Tanaka et al., 2014*), share orthologs with *U. bromivora* (*Figure 10A*). The amino acid similarity between each *U. maydis* effector and its corresponding *U. bromivora* ortholog ranges from 41% for the organ specific effector See1 to 70% for the widely conserved peroxidase inhibitor effector Pep1 (*Figure 10A*). To experimentally test if *U. bromivora* effectors can functionally complement their respective orthologs in the *U. maydis / Zea mays* pathosystem, we chose two conserved effectors for that assay, Stp1 and Pep1 (*Doehlemann et al., 2009*; *Schipper, 2009*). These effectors were recently shown to play an essential role for virulence of *U. maydis*. The avirulent phenotype of both *U. maydis* deletion mutants could be complemented by introducing the corresponding *U. bromivora* ortholog (*Figure 10B*). This demonstrates that the *U. bromivora / Brachypodium* system could be indeed suitable to study the functional role and host targets of conserved effectors from related species.

Besides the described orthologs, we found 427 orphan *U. bromivora* genes that lack orthologs in the five other fungi they were compared with. Of these, 21 are predicted to encode secreted proteins. None of the orphan genes had a predicted function according to either FunCat or Blast2GO.

## Transposable elements and genome defense machinery

Comparing the currently available smut genomes both, *U. hordei* and, to an even greater extent, *U. bromivora* contain a high number of transposable elements and other repetitive sequences, encompassing up to 14.33% of the genome (*Table 2*). Similar to the other smuts sequenced to date, the *U. bromivora* genome has a low frequency of small interspersed nuclear elements (SINEs) (0.18% of the genome), a class of retrotransposons that lost the coding region for their own reverse transcriptase.

**Table 3.** Positive selection among sequenced smut fungi

| | U. bromivora | U. maydis | U. hordei | S. scitamineum | S. reilianum | M. pennsylvanicum |
|---|---|---|---|---|---|---|
| **Genes analysed** | | | | | | |
| Total | 4947 | 4947 | 4947 | 4947 | 4947 | 4947 |
| Under selection (q = 0.05) | 140 | 345 | 188 | 170 | 256 | 2390 |
| Under selection (q = 0.01) | 96 | 116 | 135 | 90 | 91 | 1434 |
| **Non-PSEPs** | | | | | | |
| Total | 4738 | 4735 | 4733 | 4748 | 4743 | 4753 |
| Under selection (q = 0.05) | 128 | 318 | 168 | 153 | 240 | 2287 |
| Under selection (q = 0.01) | 86 | 105 | 122 | 81 | 84 | 1377 |
| **PSEPs** | | | | | | |
| Total | 209 | 212 | 214 | 199 | 204 | 194 |
| Under selection (q = 0.05) | 12 | 27 | 20 | 17 | 16 | 103 |
| Under selection (q = 0.01) | 10 | 11 | 13 | 9 | 7 | 57 |

*Number of genes under positive selection in each of the six fungi used in this study. The number of genes predicted to be under positive selection is given for both a FDR of 0.05 and 0.01 and are grouped in three categories: total genes analyzed, the subset of genes that are not predicted to be secreted and the subset of genes that are predicted to be secreted. This analysis is limited to the 4,947 one-to-one orthologs that were used for the construction of the phylogeny.

However, it contains a high percentage of long terminal repeat (LTR) retrotransposons (5.83%), which are independent from other mobile genomic elements and encode all proteins necessary for transposition. Similar ratios of long interspersed elements (LINE) have spread in the *U. hordei* and the *U. bromivora* genome (4.62% and 4.38%, respectively; *Table 2*). With few exceptions, such as the mating type chromosome (Chr1), transposable elements are evenly distributed over the *U. bromivora* chromosomes (*Figure 7—figure supplement 1*). These results are in line with those made in *U. hordei*, where except for the mating type region, repetitive sequences are also evenly distributed across the genome (*Laurie et al., 2012*; *Bakkeren et al., 2006*).

While transposable elements have clearly shaped the genome of *U. bromivora*, their action might be counter-balanced by the presence of a functional core machinery of the RNA interference pathway. Homology searches identified *UBRO_08937* that likely encodes the PAZ-domain (Piwi/Argonaute/Zwille-domain) containing endoribonuclease DICER. *UBRO_20628*, *UBRO_01631*, and *UBRO_08874* encode three RNA-dependent RNA Polymerases (RdRP) which are necessary for the formation of the complementary strands of target RNA, and therefore represent a prerequisite for RNA-directed silencing. *UBRO_06256* is predicted to encode the argonaute protein, the catalytic subunit of the RISC-complex. We also identified the small RNA methyltransferase Hen1 (UBRO_08578). Moreover, the chromodomain (CD) protein CHP1 and CHP2 important for heterochromatic gene silencing are with UBRO_05116 and UBRO_07750 as well present. Analysis of dinucleotide frequencies in repetitive regions of the genome shows a lack of CpG dinucleotides similar to that observed in *U. hordei* (*Laurie et al., 2012*) (*Figure 11*). This may indicate the presence of repeat induced point-mutations (RIP) which can serve as an additional genomic defense mechanism against transposable elements (*Selker, 2002*).

## Genes under positive selection in *U. bromivora*

Genes under positive selection are indicative of an ongoing adaptation process often found either upon a host jump, neofunctionalization of a protein, or during the arms race between the host and the pathogen. We analysed the proteins of our six fungal genomes for signs of positive selection. Our analysis of the one-to-one orthologs between the six smut fungi we compared (*Table 3*) confirmed the previously reported finding of high levels of positive selection in *M. pennsylvanicum* (*Sharma et al., 2014*). It also showed that *U. bromivora* has the lowest levels of selection when measured at a false discovery rate (FDR) of 0.05 but is between *U.maydis* and *S. reillianum* when using an FDR of 0.01. Among the genes being under positive selection, only twelve encode predicted

secreted proteins. In contrast to the twelve genes found in *U. bromivora*, *M. pennsylvanicum*, which adapts to its dicot host, harbors 103 genes encoding predicted secreted proteins that were shown to be under positive selection. The evolutionary pressures driving the observed incidents of selection in *U. bromivora* remain unclear.

## Conclusion

Smut fungi, especially *U. maydis*, have become models for studying recombination (*Holliday, 1964*), cell biology (*Steinberg et al., 2008*; *Haag et al., 2015*), and, due to their nature, biotrophic interactions (*Brefort et al., 2009*). This has resulted in *U. maydis* being considered as an important model pathogen in the scientific community (*Dean et al., 2012*). In recent years, the importance of small secreted molecules termed effectors, which shape the biotrophic interaction between the pathogen and its host, has become increasingly evident. One major challenge for effector research is that most effector proteins have no sequence similarity to any known protein and are therefore difficult to functionally characterize. Although these important molecules are produced by the pathogen, in many cases they target host processes and therefore require a completely accessible host system for complementary functional studies in both the host and pathogen. In our search to identify a biotrophic model pair for a smut and a temperate host grass that due to their genetic accessibility will enable these complementary functional studies, we chose the smut fungus *U. bromivora* and its compatible host grass *Brachypodium* sp.

Although *U. bromivora* is closely related to other sequenced model smuts, it displays interesting peculiarities in its lifestyle, especially in connection with its mating system. Most strikingly is one major feature: the mating type bias. Upon spore germination and meiosis, the *a* and *b* loci located on chromosome 1 co-segregate, and progeny with two different mating types arise from one spore, MAT-1 and MAT-2. Interestingly, the locus that causes the mating type bias co-segregates with the MAT-2 mating type region leading to the inability to survive under saprotrophic conditions. Independent of its cause, it has important implications for the biology of *U. bromivora*. The bipolar mating system and the intratetrad mating entail a strong tendency for inbreeding which could be an advantageous driving speciation of this highly specialized plant pathogen (*Hoekstra, 2005*). This holds especially true as the pheromone receptor based cell-cell recognition system is rather promiscuous between different smut species and might not be sufficient as a speciation barrier (*Kellner et al., 2011*). Besides the removal of detrimental DNA, like transposable elements and deleterious mutations, sexual reproduction is a way to efficiently reshuffle alleles over generations to provide an opportunity for natural selection to produce efficient allele combinations (*Goddard et al., 2005*; *Lee et al., 2010*). The relative abundance of transposable elements in *U. bromivora* could potentially serve as a source of variation to counterbalance the assumed loss of heterozygosity due to inbreeding between progeny from the same spore. This could enable the population to adapt to evolving challenges such as host defense mechanisms. As a consequence, it might be essential to keep the functional machinery for RNA silencing intact to limit the uncontrolled spreading of transposable elements with potentially deleterious effects. In contrast to *U. bromivora*, *U. maydis* has evolved a very efficient recombination system enabling the removal of most of its invasive transposable elements. This was likely a prerequisite to allow the loss of the silencing machinery in *U. maydis*, which subsequently led to the gain of a competitive advantage through symbiosis with double-stranded RNA totiviruses, which encode for killer toxins that target non-killer containing competitive microbes (*Drinnenberg et al., 2011*; *Koltin and Day, 1976*).

The deleterious allele which leads to the mating type bias in *U. bromivora* is still unidentified. In the related smut *Microbotryum violaceum* (formerly described as *Ustilago violacea*), intratetrad mating was observed as a result of a recessive haplo-lethal allele (*Hood and Antonovics, 2000*). In *Ustilago nuda*, a recessive proline biosynthesis allele linked to the mating type locus led to a similar result in the haploid stage of the dimorphic life cycle (*Nielsen, 1968*). Alternatively, a dominant mechanism via meiotic drive elements as described for *Neurospora crassa* (*Turner and Perkins, 1979*) and other ascomycetes such as *Gibberella fujikuroi* (*Fusarium verticillioides*) (*Kathariou and Spieth, 1982*) and *Podospora anserina* (*Bernet, 1967*) could be causative for the observed mating type bias in *U. bromivora*. While, in this scenario, the 'killer'-allele and 'resistance'-allele would form a locus and would be linked to the recombination-suppressed MAT-1 region, the corresponding 'Non-killer/susceptibility' alleles would co-segregate with the MAT-2 region. The alleles of the MAT-

2 strain would therefore lead to its haplo-lethality. Future research will clarify the cause of the mating type bias in *U. bromivora*.

In summary, the newly established model system *U. bromivora* and *Brachypodium* sp. has tremendous potential for the study of biotrophy related questions on both the pathogen and host side as well as evolutionary questions such as sex and speciation. The high quality, manually curated fungal genome, available RNA-seq data, and growth as well as transformation protocols for both the host and the pathogen, provide a solid basis for scientists to gain new insights into biotrophic plant pathogen interactions.

## Materials and methods

### Strains, plasmids, and fungal culture conditions

DNA manipulation and plasmid generation were performed according to standard molecular cloning procedures (*Chong, 2001*; *Ausubel et al., 1987*). All DNA manipulations were performed with *E. coli* MACH1 (Thermo Fisher Scientific, Waltham, MA). Primers and plasmids are compiled in *Supplementary file 1*. All sequences of plasmids created in this study are provided as gb/gbk files in *Supplementary file 2*. Strains used in this study are listed in *Supplementary file 1*. The *Ustilago bromivora* spore material used in this study was obtained from Thierry Marcel and originated from spontaneous repetitive infections which occurred in a greenhouse at INRA UMR BIOGER, Avenue Lucien Brétignières BP01, 78850 Thiverval-Grignon, France (*Barbieri et al., 2012*). *U. bromivora* UB1 (formerly named UB2112) and UB2 were cultivated in Potato dextrose (PD) liquid medium (2.4% PD dissolved in $H_2O$; Becton, Dickinson and Company, Franklin Lakes, New Jersey) at 21°C, shaking at 200 rpm in baffled flasks. *U. maydis* and *U. hordei* were cultivated according to *Kämper et al. (2006)* and *Laurie et al. (2012)*. *U. maydis* strains were generated by gene replacement via homologous recombination as described by *Kämper (2004)* or by insertion of p123 derivatives into the *ip* locus (*Loubradou et al., 2001*).

### *U. maydis* virulence assays

*U. maydis* virulence assays were performed as described by *Kämper et al. (2006)*. In brief, the solo-pathogenic strain SG200 and its derivatives were cultivated in $Yeps_{Light}$ (0.4% yeast extract, 0.4% peptone, 2% sucrose) at 28°C, under continuous shaking (200 rpm), until they reached an $OD_{600 nm}$ of 0.8. After centrifugation at 2400 g for 5 min, cultures were adjusted in $H_2O_{dd}$ to an $OD_{600 nm} = 1$. The suspensions were subsequently used for syringe-inoculation of seven day old maize seedlings (variety Early Golden Bantam). Infection symptoms were scored twelve days post infection employing the scoring system described by *Kämper et al. (2006)*.

### Genomic DNA extraction for Single Molecule Real-Time (PacBio) and Illumina sequencing

Genomic DNA (gDNA) extraction was performed as previously described for *U. maydis* (*Kämper et al., 2006*). *U. bromivora* cultures were grown to an exponential phase and subjected to Phenol-Chloroform extraction. For Single Molecule Real-Time (PacBio) sequencing of UB1, Phenol-Chloroform extraction was followed by an additional purification step via the Power Clean DNA Kit (MO BIO Laboratories, Carlsbad, CA). For Illumina sequencing of UB2, Phenol-Chloroform extracted gDNA was purified using the MasterPure Complete DNA and RNA Purification Kit (Epicentre, Madison, WI).

### Library preparation and Single Molecule Real-Time (PacBio) sequencing

The SMRT bell was produced using the DNA Template Prep Kit 1.0 (Pacific Biosciences, Menlo Park, CA). The input genomic DNA concentration was measured using a Qubit Fluorometer dsDNA Broad Range assay (Thermo Fisher Scientific, Waltham, MA). 10 μg of gDNA was mechanically sheared to an average size distribution of 15 kb, using a Covaris gTube (Kbiosciences, Hoddesdon, UK). A Bioanalyzer 2100 12K DNA Chip assay (Agilent, Santa Clara, CA) was used to assess the fragment size distribution. 5 μg of sheared gDNA was DNA damage repaired and end-repaired using polishing enzymes. A blunt end ligation reaction followed by exonuclease treatment was performed to create the SMRT bell template. A Blue Pippin device (Sage Science, Beverly, MA) was used to size select

the SMRT bell template and enrich for large fragments > 10 kb. The size selected library was quality inspected and quantified on an Agilent Bioanalyzer 12 kb DNA Chip and on a Qubit Fluorimeter (Thermo Fisher Scientific, Waltham, MA), respectively.

A ready to sequence SMRT bell-Polymerase Complex was created using the P6 DNA/Polymerase binding kit 2.0 (Pacific Biosciences, Menlo Park, CA) according to the manufacturer's instructions.

The Pacific Biosciences RS2 instrument was programmed to load and sequence the sample on 5 SMRT cells (v3.0; Pacific Biosciences, Menlo Park, CA), taking 1 movie of 240 min each per SMRT cell. A MagBead loading (Pacific Bioscience, Menlo Park, CA) method was chosen in order to improve the enrichment of longer fragments.

After the run, a sequencing report was generated for every cell via the SMRT portal, in order to assess the adapter dimer contamination, the sample loading efficiency, the obtained average read-length, and the number of filtered sub-reads.

## UB1 de novo genome assembly

The genome was assembled with Pacific Bioscience's SMRTanalysis software version 2.2.0 and the hierarchical genome-assembly process (HGAP) v3 protocol (*Chin et al., 2013*) including polishing with Quiver. Default settings were used, except for selecting only reads with a minimum length of 10,000 bp and read quality above 0.8. Assembly and polishing resulted in 25 contigs with an overall length of 20.7 Mb.

## Transcriptome assembly

*U. bromivora* UB1 was grown in axenic culture (21°C, 200 rpm, PD medium) in three independent biological replicates to an exponential phase ($OD_{600\ nm}$ = 0.8). RNA was extracted using the TRIzol method (*Chomczynski and Sacchi, 2006*) according to the manufacturer's protocol (Thermo Fisher Scientific, Waltham, MA). Residual DNA was removed with the DNA-*free* Kit (Thermo Fisher Scientific, Waltham, MA). The extracted and purified RNA was used for library generation with the NEB Next Ultra RNA Library Prep Kit according to the manufacturer's protocol (cutout size 200–800 bp; New England Biolabs, Ipswich, MA) and was sequenced with an Illumina HiSeq2000 instrument in paired-end 100 mode. The resulting data was pooled to yield 68M read pairs. The overall quality metrics were verified with fastqc (http://www.bioinformatics.babraham.ac.uk/projects/fastqc). The pooled reads were then assembled using Trinity (vr20140413p1) using the built-in trimmomatic quality trimming, in-silico read normalization and jaccard clipping procedures (*Grabherr et al., 2011*). The resulting transcriptome assembly consisted of 11,918 contigs with an N50 of 3,027 bp.

## Prediction of open reading frames and proteome analysis

Primary structural annotation was achieved by mapping the protein sequences of *U. hordei* on the scaffolds using exonerate (v2.2.0) unless the protein sequences could not be mapped (*Slater and Birney, 2005*). As a de novo gene predictor, GeneMark-ES version 2 was applied (*Ter-Hovhannisyan et al., 2008*). In addition, the orthologous protein sequences of *U. maydis*, *S. reilianum* and *U. hordei* were inspected by multi T-Coffee (v8.69) alignments to further validate the gene structure in *U. bromivora* (*Notredame et al., 2000*). As transcriptional evidences, RNA-seq reads were mapped on the genome using tophat2 (v2.0.8). The interval for allowed intron lengths was set to a minimum of 20 nt and a maximum of 1 kb (*Langmead et al., 2009*; *Trapnell et al., 2012*). The Trinity assembled RNA reads were mapped as transcripts (*Grabherr et al., 2011*). The different gene structures and supporting evidence were displayed in GBrowse (*Donlin, 2009*), allowing manual validation and correction of all coding sequences. The final call set comprises 7,233 protein coding genes. In addition, 133 tRNA-encoding genes are predicted using tRNAscan-SE (*Lowe and Eddy, 1997*). The protein coding genes were analyzed and functionally annotated using the PEDANT system (*Walter et al., 2009*), accessible at http://pedant.helmholtz-muenchen.de/genomes.jsp?category=fungal. The genome and annotation were submitted to the European Nucleotide Archive (http://www.ebi.ac.uk/ena) under the study number PRJEB7751.

The predicted protein set was searched for highly conserved single (low) copy genes to assess the completeness of the sequence dataset. Orthologous genes to all 246 single copy genes were identified by BLASTP comparisons (eValue: $10^{-3}$) against the single-copy families from all 21 species available from the FunyBASE (*Aguileta et al., 2008*). Additionally, 247 of 248 core-genes commonly

present in higher eukaryotes (CEGs) could be identified by BLASTP comparisons (eValue: $10^{-3}$) (*Parra et al., 2009*).

## Differential expression and overrepresentation analyses in the transcriptomic dataset

For differential expression analysis, RNA from axenic culture of UB1 (see 'Transcriptome assembly') and from seedlings, 12 days after planting of germinating caryopses that were incubated with spore material for 1 week at 4°C, was isolated using the TRIzol method and DNA was removed with the DNA-*free* Kit (Thermo Fisher Scientific, Waltham, MA). After library preparation (cutout size: 200–800 bp; NEB Next Ultra RNA Library Prep Kit; New England Biolabs, Ipswich, MA) three independent biological replicates were sequenced with an Illumina HiSeq2000 instrument, paired-end 100 bp.

RNA-Seq was quantified against the combined transcriptome extracted from *Brachypodium distachyon* Bd21 (Bdistachyon_283_v2.1) and *Ustilago bromivora* UB1 annotations with kallisto using default parameters (*Bray et al., 2016*). In our dataset we identified 20 cases, where more than one splicing variant encoded by the same gene was present. In all subsequent analyses, splicing variants were treated and counted as individual genes. Differential expression statistics between axenic and *in planta* samples were computed using the DeSeq2 R package under the assumption that the overall expression level is similar between the two samples (*Love et al., 2014*). Transcripts were considered significantly up- or downregulated *in planta*, if the log2fold-change compared to axenic culture was $\geq 2/\leq -2$ and the Benjamini-Hochberg (*Hochberg and Benjamini, 1990*) corrected p-value was $\leq 0.1$. Over-/underrepresentation of individual functional classes of interest (e.g. predicted secreted proteins) among the *in planta* up- and downregulated transcripts was tested by Fisher exact test in the R statistical environment (*Core Team, R, 2011*). Systematic over-/underrepresentation analysis for all functional classes present in the FunCat annotation of the given dataset was conducted with the FunCat workflow (*Ruepp et al., 2004*). Expression data were submitted to GeneExpressionOmnibus (http://www.ncbi.nlm.nih.gov/geo/) under the accession number GSE87751. A more detailed description of the methodology can be found on Bio-Protocol (*Czedik-Eysenberg et al., 2017*).

## Bioinformatics analysis: Secreted protein prediction, orthologs, and transposons

To predict putative secreted proteins, protein sequences were retrieved from the PEDANT 3 Genome Database and analysed for specific features that are associated with secreted proteins. We used SignalP (v4.0) (*Petersen et al., 2011*) to predict the existence of a signal peptide, TMHMM (v2.0c) (*Sonnhammer et al., 1998*; *Krogh et al., 2001*) to predict the existence of transmembrane domains, Phobius (v1.01) (*Kall et al., 2004*) to detect both signal peptides and transmembrane domains, TargetP (v1.1b) (*Emanuelsson et al., 2000*) to predict the final location of a protein, and ScanProsite (*Gattiker et al., 2002*) to detect the presence of the ER retention motif [KRHQSA]-[DENQ]-E-L. A list of putative secreted proteins was generated that met the criteria of (1) signal peptide predicted by both SignalP and Phobius, (2) fewer than two transmembrane domains predicted by both TMHMM and Phobius, (3) no ER retention motif and (4) not predicted to target the mitochondrion.

Orthologs were detected using OrthoMCL (v2.0.9) using default settings (*Fischer, 2011*). OrthoMCL takes a list of proteins as an input and was provided with the full proteomes of *U. maydis*, *U. bromivora*, *U. hordei*, *S. scitamineum*, *S. reilianum*, and *M. pennsylvanicum*. Orthologous proteins from the different genomes are then sorted into groups. OrthoMCL makes use of the MCL algorithm (*Enright et al., 2002*). Only nuclear genes were used for the prediction of orthologous relationships.

Protein alignments were performed in T-Coffee (v11.00.8cbe486) using the default settings. BLAST searches were performed with the stand-alone BLAST+ suite (*Camacho et al., 2009*) where possible. However, some programs required the older C legacy toolkit. We used RepeatScout (*Price et al., 2005*) for the de novo identification of repeat families in combination with the RepBase database (*Jurka et al., 2005*) to detect previously published transposable elements, pseudogenes, and retroviruses. The combined library of de novo and RepBase repeats were used to identify individual repeat elements on the genome using RepeatMasker (*Smit et al., 2010*). All determined

repeat elements were classified using TEclass (*Abrusan et al., 2009*) and analyzed for fingerprints of repeat-induced point mutations (RIP) regarding overrepresented dinucleotide frequencies using RIP-CAL (*Hane and Oliver, 2008*).

## Fungal phylogeny and prediction of positive selection

5,121 genes were predicted as one-to-one orthologs, that means that there was an ortholog detected in each of the six fungal species and there was only a single copy in each genome. After removal of genes that were unsuitable for the positive selection analysis like genes without one-to-one orthologs or which had more than one predicted transcript in one or more species, 4,947 sequences remained. The ORF sequences were converted to amino acid sequences with T-Coffee (v11.00.8cbe486), aligned and then converted back into nucleotide sequences. These nucleotide alignments were concatenated to form a single alignment which was used as the input for RAxML (v8.1.16) (*Stamatakis, 2014*). The tree was generated using the rapid bootstrapping and best-scoring maximum likelihood tree algorithm, GTRGAMMA nucleotide model and 1000 bootstraps.

The individual alignments and the unrooted species tree were used as inputs for the codeml program from PAML 4.8A. (*Yang, 1997*). Two control files were used, both, allowing two or more dN/dS ratios for branches, checking for positive selection and having an initial omega value of 1. One allowed the omega value to vary while the second kept it constant. The likelihood ratios under the two models were compared for each gene with the formula $\Delta LRT = 2 \times (lnL1 - lnL0)$. The resulting value could be assessed using the chi-squared distribution with 1 degree of freedom to determine if the gene was under positive selection or not. Raw p-values were adjusted to take into account the false discovery rate using the qvalue R package (http://qvalue.princeton.edu/, http://github.com/jdstorey/qvalue) (*Storey, 2015*).

## Illumina sequencing of UB2, mapping to UB1, and SNP calling

Genomic DNA of UB2 was used for library generation (insert size 600–900 bp; NEBNext Ultra DNA Library Prep Kit for Illumina; New England Biolabs, Ipswich, MA) and sequenced with an Illumina HiSeq2500 instrument in 125 bp paired-end mode. Reads were mapped to the genome of UB1 using CLC Genomics Workbench (v.7.0.3; Qiagen, Hilden, Germany) with the following parameters: mismatch cost = 2, insertion cost = 3, deletion cost = 3, length fraction = 0.5, similarity fraction = 0.8, no global alignment. SNP calling was performed via the quality-based variant detection mode with the following parameters: neighbourhood radius = 4, maximum gap and mismatch count = 2, minimum central quality = 10, non-specific matches, broken pairs and variants in non-specific regions were ignored, minimum coverage = 90, minimum variant frequency = 95%, maximum expected alleles = 2, maximum coverage = 203, sufficient variant count = 85, required variant count = 85. For calling SNPs the presence in both forward and reverse reads was a requirement. Heterozygous SNPs as well as small insertions and deletions were not considered. Moreover, the mitochondrial genome was not included in the analysis.

## UB2 de novo genome assembly from Illumina paired-end reads

De novo assembly of the MAT-2 strain UB2 was performed with SOAPdenovo2 (*Luo et al., 2012*) (127mer version 1.4.10) with kmer lengths ranging from 43 to 115 in steps of 6 with the following parameters: max_rd_len = 120, avg_ins = 470, asm_flags = 3, rd_len_cutoff = 120, rank = 1, pair_num_cutoff = 3, map_len = 32. While assemblies with kmer lengths 73–91 performed good at diverse metrics, kmer 91, at 4,712 bp and 8,160 bp, had the best mean size for both contigs and scaffolds, respectively, and the highest number of contigs larger than 100 kb. The resulting assembly (kmer 91) had a contig and scaffold N50 of 23,058 bp and 113,827 bp. Scaffolds obtained after de novo assembly as well as the raw sequencing reads were submitted to the European Nucleotide Archive (http://www.ebi.ac.uk/ena) under the study number PRJEB7751.

## Spore recovery and sterilisation

Infected spikelets were dehusked and ground with a pistil in a 1.5 ml microcentrifuge tube. After the addition of 250 µl water, grinding continued until a black spore suspension was visible. The spore suspension was subjected to $CuSO_4$ treatment to kill vegetative fungal cells and bacteria. $CuSO_4$ was added to the suspension at a final concentration of 1.5%, solution and spores were incubated

for 15 min, and washed 3 times with water to remove all traces of $CuSO_4$. $CuSO_4$-treated spore material was resuspended either in 500 µl of water supplemented with 100 µg ml$^{-1}$ ampicillin, 50 µg ml$^{-1}$ tetracycline, and 25 µg ml$^{-1}$ chloramphenicol and plated on PD agarose plates (2.4% PD supplemented with 2% agarose) in serial dilutions for the isolation of haploid progeny or resuspendend in 1 ml of the above-mentioned antibiotic solution and spotted on objective slides for microscopy. A more detailed description of this method can be found on Bio-Protocol (*Bosch and Djamei, 2017*).

## WGA, FM4-64, and DAPI staining

Cell walls of fungal hyphae were stained with the chitin specific Wheat Germ Agglutinin - Alexa Fluor 488 conjugate dye (WGA-AF 488; Thermo Fisher Scientific, Waltham, MA). Plant membranes were stained with FM4-64 (Thermo Fisher Scientific, Waltham, MA). Staining and confocal laser scanning microscopy was performed as previously described by *Doehlemann et al. (2008)*. For DAPI and WGA staining of germinating spores and sporidia, germinating spores/sporidia were pelleted and fixed by incubating them with acetone for 15 min. Fixation was followed by 10 min incubation with WGA-AF 488 to visualize fungal cell walls and 15 min incubation with 1 µg ml$^{-1}$ DAPI solution (Sigma-Aldrich, Taufkirchen, Germany) to stain nucleic acid. Cell/spore pellets were resuspended in PBS and subjected to confocal laser scanning microscopy. Images were acquired with a LSM780 Axio Observer confocal laser scanning microscope (Zeiss, Jena, Germany). WGA-AF excitation was at 488 nm and detection at 517–552 nm, DAPI excitation at 405 nm and detection at 421–508 nm.

## Monitoring of germination by microscopy

$CuSO_4$-treated spore suspensions were spotted on objective slides (µm dish 35 mm; IBIDI, Planegg, Germany). After freezing them for 20 min on a metal block to avoid spreading of the liquid, they were covered with a thin (<2 mm) layer of PD agar or 1.5% water agar, respectively. After a 15–17 hr incubation at 21°C, the picture acquisition was performed with an inverted microscope (Zeiss Axiovert 200 M with 40×/1.3 Plan-Neofluar Oil objective). Picture acquisition occurred every 20 min. Image processing was done with Fiji Imaging software.

## Transformation of *U. bromivora*

The transformation protocol was adapted from *Schulz et al. (1990)* and *Gillissen et al. (1992)*. *U. bromivora* cells were grown to an $OD_{600 nm}$ of 0.3–0.6. Cells were harvested at 2400 g (10 min, RT), and washed once with $Mg^{2+}$-MES buffer (20 mM MES buffer, pH 5.8, 1 M $MgSO_4$). The Pellet was resuspended in 1 ml $Mg^{2+}$-MES buffer containing 10 mg ml$^{-1}$ Glucanex (Sigma-Aldrich, Taufkirchen, Germany) and 5 mg ml$^{-1}$ Yatalase (Takara Bio, Saint-Germain-en-Laye, France) and kept on ice. Protoplastation was monitored and stopped by addition of 10 ml ice-cold $Mg^{2+}$-MES buffer when 30–40% of the cells appeared round due to the loss of the cell wall. Cells were washed 3 times (2400 g, 10 min, 4°C) in $Mg^{2+}$-MES and once in STC buffer (STC: 100 mM $CaCl_2$, 10 mM Tris-HCL pH 7.5, 0.9 M sorbitol). Protoplastation was followed by PEG-mediated transformation. To this end, 5 µg plasmid DNA and 1 µl 100 mg ml$^{-1}$ Heparin were added to 100 µl protoplasts and incubated for 30 min on ice. 500 µl STC-PEG (40% PEG4000 in STC buffer) were added and mixed by pipetting gently up and down. After 15 min incubation on ice, the mixture was plated on PD regeneration agar plates composed of an lower layer of PD regeneration agar with twice the concentration of antibiotic (2.4% PD, 0.9 M sorbitol, 1.5% agar supplemented with either 4 µg ml$^{-1}$ Carboxin, 200 µg ml$^{-1}$ Geneticin G418, or 200 µg ml$^{-1}$ Hygromycin B) overlaid by an antibiotic-free layer of PD regeneration agar. Colonies were visible after 10 to 14 days. For stable genomic integrations, restriction enzyme mediated transformation was performed according to *Bölker et al. (1995)*. In brief, 25 U of the restriction enzyme XbaI and XbaI-linearized p123UB-GFP were added to protoplasts and the aforementioned transformation protocol was followed immediately after addition of enzyme and plasmid.

## Plant growth conditions and sexual propagation

Caryopses of *B. distachyon* accession Bd21 were kindly provided by Phillipe Vain (*Vain et al., 2008*), caryopses of accession ABR4 by John Vogel. For production of donor material, caryopses of each accession were gas-sterilized and transferred to Ø = 10 cm pots with a 4:1 mixture of standard potting soil (Einheitserde Werkverband e.V., Sinntal-Altengronau, Germany) and perlite (Granuflor, Vechta, Germany). Germination and early plant growth took place in a phyto chamber (Johnson

Controls, Milwaukee, WI) under the following conditions: 20 hr light (150 µE), 24°C; 4 hr dark, 18°C; 60% humidity. After 10 days, pots were transferred to a cold room to vernalize plants for 6 weeks at 4°C, 40 µE, 13 hr light period. Pots were then moved back to the phyto chamber to allow plants to flower. Four weeks after anthesis, plants were transferred to the greenhouse for caryopses maturation and drying. Harvested caryopses were stored at 4°C in the dark.

## Surface-sterilization of *B. distachyon* caryopses, vernalization, and infection with *U. bromivora* spores and planting

Caryopses within husks were gas sterilized (*Clough and Bent, 1998*). After sterilization, caryopses were moistened with a few ml of sterile water without submerging the caryopses and incubated in the dark at 4°C for 1 week to germinate. The seedlings were then moistened with a spore/water suspension for infection for one additional week at 4°C in the dark. Subsequently the seedlings were potted in a mixture of 3:1:1:1 standard potting soil (Einheitserde Werkverband e.V., Sinntal-Altengronau, Germany): perlite (Granuflor, Vechta, Germany): silica sand (min2C GmbH, Melk, Austria): germination soil (Neuhaus 'Huminsubstrat', Klasmann-Deilmann GmbH, Geeste, Germany), at a depth of ~1 cm, so that the emerged shoot remained exposed to the air and light. After 10 days in the growth chamber (20 hr light (150 µE), 24°C; 4 hr dark, 18°C; 60% humidity) the pots were transferred to 4°C, 40 µE, 13 hr light period for 3 weeks (Bd28) or 6 weeks (ABR4), for vernalization and subsequently returned to the growth chamber. 4–5 weeks later, the infected, spore filled spikelets were visible.

## Screen to isolate a compatible mating partner

*U. bromivora* spore material was surface sterilized and germinated on PD plates (see spore recovery and sterilization). After spore germination, all derived colonies were pooled and used as an inoculum for further proliferation in liquid PD medium for 24 hr to ensure that only strains, which are viable in axenic culture, would grow. These mixtures were used as an infection inoculum of 300 vernalized *B. hybridum* caryopses (Bd28). After six weeks the infected spikelets were harvested, derived spores were surface sterilized and plated on solid PD medium. Colonies derived from single spores were singled out and tested by diagnostic PCR for the presence of *pra2*.

## Infection of *B. hybridum* with *U. bromivora* liquid culture

To infect *B. hybridum* Bd28 with a pair of compatible mating partners, both strains were grown until they reach the exponential phase ($OD_{600 \, nm} = 0.8$) and set to an $OD_{600 \, nm} = 2$ with $H_2O_{dd}$. Both strains were mixed in equal amounts. One or two days before inoculation, vernalized *B. hybridum* caryopses were placed to RT to promote germination. Upon inoculation with the fungal mixture, the onset of coleoptiles should be visible. Seedlings were moistened with the fungal mixture and incubated in an appropriate tube, e.g. 2 ml microcentrifuge tube, at 21°C in the dark. After 24 hr incubation, seedlings were potted.

## Callus culture generation of Bd28

Mature embryos from dormant caryopses were used for callus culture, plant regeneration, and transformation. In brief, dry dormant caryopses were surface-sterilized for 45 min with 6% NaClO including 0.03% Tween20. After surface-sterilization, caryopses were rinsed five times with sterile tap water. Embryo preparation was carried out in two steps. First, the embryo was cut away from the endosperm. Second, the embryo was cut in a longitudinal direction into two pieces. Forty bisected embryos were transferred to a Ø = 10 cm petri dish with longitudinal wound in direct contact with the callus induction medium (CIM). Callus induction of *B. hybridum* Bd28 took place on CIM (4.3 g $l^{-1}$ MS No.4 (Murashige and Skoog medium modification No. 4; Duchefa Biochemie, Haarlem, The Netherlands), 30 g $l^{-1}$ maltose, 11.1 µM 2,4-D, 2 mM $NH_4NO_3$, 1.9 mM MES-monohydrate, 1x B5 vitamin mixture (Duchefa Biochemie, Haarlem, The Netherlands), 3.5 g $l^{-1}$phytagel (Sigma-Aldrich, Taufkirchen, Germany), pH 5.8. Before transformation, bisected embryos were pre-cultured for 6–8 weeks at 24°C without light. Every 14 days, calli were transferred to fresh CIM. Developing roots and shoots were cut away.

## A. tumefaciens-mediated transformation of B. hybridum Bd28

*Agrobacterium tumefaciens* strain AGL1 was used for transformation (*Lazo et al., 1991*). The plant transformation vector p6U contains a hygromycin phosphotransferase gene driven by the *Zea mays* ubiquitin promoter to confer hygromycin resistance to transformed plant cells. For *B. hybridum* Bd28 transformation, AGL1 harboring the respective p6U derivative was cultivated at 28°C and 210 rpm in Erlenmeyer flasks containing 10 ml MG/L medium (*Jones et al., 2005*) supplemented with 100 µg ml$^{-1}$ Carbenicillin, 50 µg ml$^{-1}$ Rifampicin, and 100 µg ml$^{-1}$ Spectinomycin. After 22 hr, 500 µM Acetosyringone (Sigma-Aldrich, Taufkirchen, Germany) was added and the p6U-containing AGL1 strain was cultivated for additional 2 hr. For inoculation of pre-cultured *B. hybridum* calli, the *A. tumefaciens* culture was diluted with infection solution (4 g l$^{-1}$ Chu (N6) minerals (Duchefa Biochemie, Haarlem, The Netherlands), 6.75 µM 2,4-D, 36 g l$^{-1}$ glucose, 68.4 g l$^{-1}$ sucrose, 0.7 g l$^{-1}$, 6 mM L-proline, 1x Chu (N6) vitamin mixture (Duchefa Biochemie, Haarlem, The Netherlands), 500 µM Acetosyringone (Sigma-Aldrich, Taufkirchen, Germany), pH 5.2) to OD$_{550\ nm}$ = 0.8.

6–8 weeks after callus induction, forty calli were transferred into 15 ml tubes and inoculated with *A. tumefaciens* suspension. After 30 min, the *A. tumefaciens* suspension was discarded, calli were transferred to sterile filter paper and dried for 30 min. Calli were then transferred to *Co-culture-medium* (2 g l$^{-1}$ Chu (N6) minerals, 34.2 g l$^{-1}$ sucrose, 2 mM CaCl$_2$, 20 µM Dicamba, 25 mM L-proline, 2.3 mM MES-monohydrate, 3.3 mM L-cysteine, 1x B5 vitamin mixture (Duchefa Biochemie, Haarlem, The Netherlands), 500 µM Acetosyringone (Sigma-Aldrich, Taufkirchen, Germany), 3.5 g l$^{-1}$ phytagel, pH 5.8) and co-cultivated for 3 days, at 21°C without light. After co-cultivation, calli were transferred to CIM supplemented with 300 mg l$^{-1}$ Timentin (Duchefa Biochemie, Haarlem, The Netherlands) for a 5 days resting phase (counterselection of agrobacteria) at 24°C without light. After the resting phase, calli were transferred to CIM supplemented with 300 mg l$^{-1}$ Timentin and 50 mg l$^{-1}$ Hygromycin B (Sigma-Aldrich, Taufkirchen, Germany) for an 8–12 week selection phase at 24°C without light. During that time, calli were transferred to fresh selection medium every two weeks. After these 8–12 weeks of selection, surviving calli were transferred to regeneration medium (K4N) (*Kumlehn et al., 2006*) including 150 mg l$^{-1}$ Timentin and 25 mg l$^{-1}$ Hygromycin B and cultivated for 8–12 weeks at 25°C with a 16/8 hr (light/dark) photoperiod. Regenerating plantlets from independent calli were transferred to pots with a substrate mixture of 3:1:1:1 Einheitserde:perlite:sand and grown as described previously (see 'Plant growth conditions and sexual propagation). Cyan fluorescent protein peroxisome (eCFP-SKL) Bd28-marker lines were tested by PCR and confocal laser scanning microscopy.

## Acknowledgements

We would like to thank the GMI / IMBA / IMP service facilities as well as the Campus Scientific Facilities for excellent technical support. We would like to thank Julia Riefler, Marina Pérez and Meritxell Amigo for excellent technical support, Andrea Patrignani and the Functional Genomics Center Zürich for outstanding PacBio sequencing, Eva Stukenbrock for alerting us about the existence of *U. bromivora* as well as James Matthew Watson and Nick Fulcher for valuable input on the manuscript. We thank the Max Planck Society for their generous financial support. The research leading to these results has received funding from the European Research Council under the European Union's Seventh Framework Programme (FP7/2007–2013) / ERC grant agreement n° [EUP0012 „Effectomics"], the Austrian Science Fund (FWF): [P27429-B22, P27818-B22], and the Austrian Academy of Science (OEAW). The work conducted by the US DOE Joint Genome Institute is supported by the Office of Science of the US Department of Energy under Contract no. DE-AC02-05CH11231. FR received a scholarship of the International Max Planck Research School for Environmental, Molecular and Cellular Microbiology in Marburg.

## Additional information

### Funding

| Funder | Grant reference number | Author |
| --- | --- | --- |
| European Research Council | EUP0012 Effectomics | Franziska Rabe |

| | | Alexandra Stirnberg<br>Denise Seitner<br>Simon Uhse<br>Janos Bindics |
|---|---|---|
| Austrian Science Fund | P27429-B22 | Franziska Rabe<br>Angelika Czedik-Eysenberg |
| Austrian Academy of Sciences | | Franziska Rabe<br>Jason Bosch<br>Alexandra Stirnberg<br>Tilo Guse<br>Lisa Bauer<br>Denise Seitner<br>Fernando A Rabanal<br>Angelika Czedik-Eysenberg<br>Simon Uhse<br>Janos Bindics<br>Bianca Genenncher<br>Fernando Navarrete<br>Armin Djamei |
| Max-Planck-Gesellschaft | | Franziska Rabe<br>Gertrud Mannhaupt<br>Regine Kahmann<br>Armin Djamei |
| Austrian Science Fund | P27818-B22 | Franziska Rabe<br>Angelika Czedik-Eysenberg |
| U.S. Department of Energy | DE-AC02-05CH11231 | John P Vogel<br>Sean P Gordon |

The funders had no role in study design, data collection and interpretation, or the decision to submit the work for publication.

### Author contributions

FR, Drafting the article, Acquisition of data, Analysis and interpretation of data; JBo, AC-E, Drafting the article, Analysis and interpretation of data; AS, DS, SU, JBi, BG, FN, Revising the article, Acquisition of data; TG, LB, Acquisition of data; FAR, MM, MCW, CMKS, UG, Revising the article, Analysis and interpretation of data; RKe, JK, JPV, SPG, Revising the article, Contributed unpublished essential data or reagents; HE, Analysis and interpretation of data; TCM, Contributed unpublished essential data or reagents; GM, Revising the manuscript, Analysis and interpretation of data; RKa, Discussion of data, Revising the article; AD, Conception and design, Acquisition of data, Analysis and interpretation of data, Drafting and revising the article

### Author ORCIDs

Ronny Kellner, http://orcid.org/0000-0002-4618-0110
Mathias C Walter, http://orcid.org/0000-0003-3012-2626
Ulrich Güldener, http://orcid.org/0000-0001-5052-8610
Armin Djamei, http://orcid.org/0000-0002-8087-9566

# Additional files

### Supplementary files

• Supplementary file 1. Primers, plasmids and strains used in this study.

• Supplementary file 2. Maps of plasmids used in this study. Plasmid maps are provided as gb/gbk files.

### Major datasets

The following datasets were generated:

| | | | Database, license,<br>and accessibility |
|---|---|---|---|

| Author(s) | Year | Dataset title | Dataset URL | information |
|---|---|---|---|---|
| Güldener U | 2016 | UB1 and UB2 genome | http://www.ebi.ac.uk/ena/data/view/PRJEB7751 | Publicly available at European Nucleotide Archive (accession no: PRJEB7751) |
| Czedik-Eysenberg A | 2016 | Global transcriptional profiling of Ustilago bromivora during axenic growth and pathogenic development | https://www.ncbi.nlm.nih.gov/geo/query/acc.cgi?acc=GSE87751 | Publicly available at NCBI Gene Expression Omnibus (accession no: GSE87751) |

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
