## [Decision Letter]

Thank you for submitting your article "A complete toolset for the study of *Ustilago bromivora* and *Brachypodium* sp. as a fungal-temperate grass pathosystem" for consideration by *eLife*. Your article has been favorably evaluated by Richard Losick (Senior Editor) and three reviewers, one of whom is a member of our Board of Reviewing Editors. The following individuals involved in review of your submission have agreed to reveal their identity: Matthew Moscou (Reviewer #2); Guus Bakkeren (Reviewer #3).

The reviewers have discussed the reviews with one another and the Reviewing Editor has drafted this decision to help you prepare a revised submission.

After intensive reading by several editors and several reviewers, we all noted that there were key discussion points that need to be included in the text as are enumerated in the individual reviews. Additionally, it was felt essential that all genomic sequencing information is deposited in public repositories and cited directly within the manuscript.

Essential revisions:

1) We would expect that all de novo assemblies be submitted before publication. It would be useful to cite the accession numbers for all of these within the Methods section.

2) Please comment directly in the manuscript on the following:

A) What is the "natural" host range of Ub?

B) What is the source of the *U. bromivora* organisms? Collection site, strain #, year; are these and the *Brachypodium* varieties deposited in / available from public seed / strain banks / collections?

C) Is there any geographic relationship between the Bd28 line (Australia) and the origin of the Ub isolate? I.e., are there known (geographic) adaptations for certain isolates?

D) How was it established that there are 23 chromosomes in Ub?

E) What is the estimated recombination rate for the MAT-2 – lethal phenotype, or some estimate of recovery of MAT-2 among xx "events"? Are the Ub1 and Ub2 isolates now considered "parental strains" that allow genetic experimentation? These may be difficult to cross with other (natural) mutations to perform genetics; what's the prospect?

*Reviewer #1:*

This manuscript does a nice job of developing and introducing a new pathosystem for the study of obligate biotrophic pathogens. I have no additional comments beyond the other two reviewers.

*Reviewer #2:*

Rabe et al. describe the development of a novel fungal-grass pathosystem that includes the description and optimization of the pathogenic lifecycle, a screen of diverse host germplasm, transformation of host and pathogen, and genome sequencing and annotation of the pathogen. In addition, the plant accession utilized is currently being sequenced and characterized by the Joint Genome Institute, part of a larger sequencing initiative for several *Brachypodium* species. The work is comprehensive and experiments were carried out in a rigorous manner. The developed pathosystem looks to be a complementary system to the current maize-*Ustilago maydis* system. While this latter system is excellent in many ways, it lacks in natural variation in resistance within maize. In contrast, the *Brachypodium-U. bromivora* system exhibit natural variation in resistance, which will aid in the identification of effectors that elicit an avirulence response and permit the systematic study of the evolutionary dynamics underlying the interaction of host and pathogen. The manuscript in its present state is quite well organized.

*Reviewer #3:*

The study by Rabe and Djamei et al. is promoting the introduction of a new pathosystem consisting of a biotrophic fungus *Ustilago bromivora* (Ub) and a monocot, *Brachypodium* species that are both closely related to the well-studied corn basidiomycete smut pathogen *Ustilago maydis* (Um) [and even closer to the barley-smut, *U. hordei* (Uh)], and cereals. Though μm has been very well studied, its interaction with corn, a difficult host to study, makes that pathosystem less than ideal. Similarly, the Uh – barley system, also previously presented as a model small-grain smut/biotroph pathosystem, is not ideal since barley is difficult to manipulate, though better for genetic studies. Interestingly, Ub is shown here to have very similar draw backs as Uh in terms of length for pathogenicity assays (though Uh is easier for infection assays and to make crosses) and lower transformation efficiency; indeed, Ub is very closely related to Uh, as is shown here. Uh actually has an advantage over Ub in that it does not have a haplolethal mutation forcing inbreeding.

Here, the authors form a collaborative group that has sequenced several host *Brachypodium* species genomes, and herewith report on the generation and analysis of the Ub genome. They also present data on genetic transformation protocols for both the pathogen and the host. This study presents a nice, well-written package and this pathosystem may well become a good experimental system to supplement others, though it's too early to tell. Although *Brachypodium* is generally considered a good cereal model system, with genome sequences, genetic tools and many mutant lines available or being generated, it is apparently not as easy as claimed to perform genetics on. And shown in this paper, infection assays are still quite technical and take certainly much longer than for Um, or other pathosystems for that matter. I guess, the usefulness will depend on the questions that require answers, or vice versa, what kinds of questions this new pathosystem will allow to be addressed.

Overall, a large amount of experimentation and analysis by many contributors is presented here in a comprehensive and well-performed way; this is no surprise given the experience these authors have with the related organisms.

I have a few issues and questions that I would like addressed:

Sequences, genome etc. do not yet appear in the ENA; I assume this is a pre-requisite for publication?

What is the "natural" host range of Ub?

What is the source of the *U. bromivora* organisms? Collection site, strain #, year; are these and the *Brachypodium* varieties deposited in / available from public seed / strain banks / collections?

Is there any geographic relationship between the Bd28 line (Australia) and the origin of the Ub isolate? I.e., are there known (geographic) adaptations for certain isolates?

Subsection “Differential expression and overrepresentation analyses in the transcriptomic dataset”: "twelve days after potting" to obtain mixed infection material should be explained better. Later on it becomes clear how spores are added to seedlings, and days after potting refers to the start of infection…?

Subsection “Fungal phylogeny and prediction of positive selection”: Please, clarify "After removal of genes that were unsuitable for the positive selection analysis".

Subsection “*A. tumefaciens*-mediated transformation of *B. hybridum* Bd28”: "Finally", what is a typical time period for such transfers before regeneration is started?

Subsection “The genome of U. bromivora”, first paragraph: How was it established that there are 23 chromosomes in Ub?

Subsection “The majority of putatively secreted proteins are of unknown function and their transcripts are overrepresented in planta”: "and therefore of unknown function", double statement with the before.

Subsection “The majority of putative *U. bromivora* effectors are not clustered”: since Ub is more related to *U. hordei*, it would make sense to include a (3-way) comparison of this important cluster; this may illustrate some of the evolutionary pressures during speciation, since this cluster harbors effectors important for plant interaction.

What is the estimated recombination rate for the MAT-2 – lethal phenotype, or some estimate of recovery of MAT-2 among xx "events"? Are the Ub1 and Ub2 isolates now considered "parental strains" that allow genetic experimentation? These may be difficult to cross with other (natural) mutations to perform genetics; what's the prospect?

Was rescue of MAT-2 haploid progeny on various supplemented media attempted? Some *Ustilago* species have a pan-deficiency for example.

Does the genome information of both mating types allow for the identification of the "lethal" (mutant?) gene? This may need sequencing of dikaryotic parental teliosore gDNA followed by a comparison to the MAT-2 recombinant progeny? Since there are only about 1300 SNPs, this may be feasible, followed by gene complementation. Some speculation appears at the end of the conclusion section.

A seemingly very low frequency of integrative transformation could be an impediment but may not have a structural basis; in *Ustilago*, a difference of 100-1000-fold is seen between transformation efficiencies with episomal constructs versus integrative ones. Improving transformation efficiencies will likely lead to better integration frequencies as well.

---

## [Author Response]

[…]

*After intensive reading by several editors and several reviewers, we all noted that there were key discussion points that need to be included in the text as are enumerated in the individual reviews. Additionally, it was felt essential that all genomic sequencing information is deposited in public repositories and cited directly within the manuscript.*

*Essential revisions:*

*1) We would expect that all* de novo *assemblies be submitted before publication. It would be useful to cite the accession numbers for all of these within the Methods section.*

The de-novo assemblies of the UB1 and UB2 as well as the raw sequencing data for UB2 have been submitted to the European Nucleotide Archive under the study number PRJEB7751. The study number has been placed into the respective Methods sections of the manuscript.

Furthermore, the RNAseq data have been submitted to GeneExpressionOmnibus under the accession number GSE87751. The accession number is now also mentioned in the respective Methods section.

*2) Please comment directly in the manuscript on the following:*

*A) What is the "natural" host range of Ub?*

In reports about *Ustilago bromivora,* infections have been observed in a broad range of Poaceae species including various *Bromus* and *Brachypodium* species [1-3]. In our manuscript we describe spore infections of *U. bromivora* on *Brachypodium distachyon, Brachypodium stacei* as well as on *Brachypodium hybridum* with full completion of the life cycle (Table 1). We assayed also infections of barley (Hordeum vulgare cv. Golden Promise) but could not observe any smut symptoms in the spikelets (unpublished results).

The following sentence has been added to the manuscript:

“The natural host range of *U. bromivora* comprises various species of the genera *Agropyron, Austrofestuca, Brachypodium including B. distachyon, Bromus, Critesion, Elymus, Festuca, Hordeum, Lolium, Sitanion and Trachynia*[1-3]. Our experiments can confirm at least three host species, *B. distachyon, B. hybridum* and *B. stacei*.”

*B) What is the source of the U. bromivora organisms? Collection site, strain #, year; are these and the Brachypodium varieties deposited in / available from public seed / strain banks / collections?*

Beside the previous reference in the Introduction: "…that the *U. maydis*-relative *U. bromivora* can infect *Brachypodium* sp.…[4]."

We added the following statement to the Methods part of the paper:

“The *Ustilago bromivora* spore material used in this study was obtained from Thierry Marcel and originated from spontaneous repetitive infections which occurred in a greenhouse at INRA UMR BIOGER, Avenue Lucien Brétignières BP01, 78850 Thiverval-Grignon, France [5].”

*C) Is there any geographic relationship between the Bd28 line (Australia) and the origin of the Ub isolate? I.e., are there known (geographic) adaptations for certain isolates?*

We do not know about any geographic adaptations of *U. bromivora* but concerning our infection assays we can state that *U. bromivora* spore infections were successful on *Brachypodium* sp. accessions originating from Asia, Africa, Australia, Europe and South America. This is also listed in Table 1 and we added the following:

“The finding that susceptible host plant accessions originate from Europe, Asia, Africa, Australia as well as South America (Table 1) underlines that *Ustilago bromivora* is considered as a cosmopolite [6].”

*D) How was it established that there are 23 chromosomes in Ub?*

The assembled contigs were compared with the synthetic chromosomes of *Ustilago hordei* which had been previously confirmed in number by optical mapping [7]. Since the *U. bromivora* contigs showed synteny with the *U. hordei* chromosomes, they were numbered as chromosome 1-23. This has been now further stated in the first paragraph of the subsection “The genome of *U. bromivora*”.

*E) What is the estimated recombination rate for the MAT-2 – lethal phenotype, or some estimate of recovery of MAT-2 among xx "events"?*

This cannot be estimated from the experimental data obtained as in a first step of the screen for a compatible MAT-2 strain, thousands of tetrads grown on plate were pooled for an infection assay. This pooled assay enriched for MAT-2 strains that were viable on plate. From the derived infected plants, spores were plated in a second round on plates and screened by PCR for viable MAT-2 strains.

The screen has been described in the subsection “Isolation of a haplo-viable *U. bromivora* MAT-2 strain” and in the subsection “Screen to isolate a compatible mating partner”.

*Are the Ub1 and Ub2 isolates now considered "parental strains" that allow genetic experimentation?*

UB1 and UB2 are not derived from the same spore tetrad – nevertheless they can mate and generate viable progeny. This we exemplified by crossing a transgenic UB1 GFP expressing strain with UB2. One can do genetic experimentation with the two strains and also use them for infection assays. See Figure 5—figure supplement 2. Please see also our clarification in the subsection “Isolation of a haplo-viable *U. bromivora* MAT-2 strain”.

*These may be difficult to cross with other (natural) mutations to perform genetics; what's the prospect?*

We do not see any general constrains why the obtained UB1 and UB2 strains should not cross with other compatible *Ustilago bromivora* isolates that are obtained independently.

Reviewer #3:

[…]

*Here, the authors form a collaborative group that has sequenced several host Brachypodium species genomes, and herewith report on the generation and analysis of the Ub genome. They also present data on genetic transformation protocols for both the pathogen and the host. This study presents a nice, well-written package and this pathosystem may well become a good experimental system to supplement others, though it's too early to tell. Although Brachypodium is generally considered a good cereal model system, with genome sequences, genetic tools and many mutant lines available or being generated, it is apparently not as easy as claimed to perform genetics on. And shown in this paper, infection assays are still quite technical and take certainly much longer than for Um, or other pathosystems for that matter. I guess, the usefulness will depend on the questions that require answers, or vice versa, what kinds of questions this new pathosystem will allow to be addressed.*

*Overall, a large amount of experimentation and analysis by many contributors is presented here in a comprehensive and well-performed way; this is no surprise given the experience these authors have with the related organisms.*

*I have a few issues and questions that I would like addressed:*

*Sequences, genome etc. do not yet appear in the ENA; I assume this is a pre-requisite for publication?*

We agree and all de novo assemblies have been submitted and identifiers have been listed in the Methods part (see above).

*What is the "natural" host range of Ub?*

Please see our response above.

*What is the source of the U. bromivora organisms? Collection site, strain #, year; are these and the Brachypodium varieties deposited in / available from public seed / strain banks / collections?*

*Is there any geographic relationship between the Bd28 line (Australia) and the origin of the Ub isolate? I.e., are there known (geographic) adaptations for certain isolates?*

Please see our response above.

*Subsection “Differential expression and overrepresentation analyses in the transcriptomic dataset”: "twelve days after potting" to obtain mixed infection material should be explained better. Later on it becomes clear how spores are added to seedlings, and days after potting refers to the start of infection…?*

It has been modified for better understanding. See the first paragraph of the subsection “Differential expression and overrepresentation analyses in the transcriptomic dataset”.

*Subsection “Fungal phylogeny and prediction of positive selection”: Please, clarify "After removal of genes that were unsuitable for the positive selection analysis".*

The genes that were removed fell into 2 categories:

1) Not one-to-one orthologs;

2) Had more than one predicted transcript in one or more species.

The text has been modified accordingly – see the last paragraph of the subsection “Fungal phylogeny and prediction of positive selection”.

*Subsection “A. tumefaciens-mediated transformation of B. hybridum Bd28”: "Finally", what is a typical time period for such transfers before regeneration is started?*

Typically, the time period for selection is 8 to 12 weeks followed by a regeneration phase of 8 to 12 weeks. Depending on the construct used for transformation these time periods can vary slightly. Therefore, we give an approximate number.

*Subsection “The genome of U. bromivora”, first paragraph: How was it established that there are 23 chromosomes in Ub?*

Please see our response above.

*Subsection “The majority of putatively secreted proteins are of unknown function and their transcripts are overrepresented in planta”: "and therefore of unknown function", double statement with the before.*

Thanks – the double statement has been removed.

*Subsection “The majority of putative U. bromivora effectors are not clustered”: since Ub is more related to U. hordei, it would make sense to include a (3-way) comparison of this important cluster; this may illustrate some of the evolutionary pressures during speciation, since this cluster harbors effectors important for plant interaction.*

We performed such a 3-way comparison of this effector cluster including now *U. hordei* and modified the text accordingly (subsection “The majority of putative *U. bromivora* effectors are not clustered”).

*What is the estimated recombination rate for the MAT-2 – lethal phenotype, or some estimate of recovery of MAT-2 among xx "events"? Are the Ub1 and Ub2 isolates now considered "parental strains" that allow genetic experimentation? These may be difficult to cross with other (natural) mutations to perform genetics; what's the prospect?*

Please see our responses above.

*Was rescue of MAT-2 haploid progeny on various supplemented media attempted? Some Ustilago species have a pan-deficiency for example.*

We attempted the rescue of MAT-2 haploid progeny on complete media that should contain all required nutrients in sufficient quantity. Moreover, we tested rescue of the second mating partner on complete medium supplemented with excess of proline, since a mating type bias due to proline-deficiency was reported for *Ustilago nuda*[8]. However, even by adding proline no MAT-2 strain could be obtained.

*Does the genome information of both mating types allow for the identification of the "lethal" (mutant?) gene? This may need sequencing of dikaryotic parental teliosore gDNA followed by a comparison to the MAT-2 recombinant progeny? Since there are only about 1300 SNPs, this may be feasible, followed by gene complementation. Some speculation appears at the end of the conclusion section.*

The identification of the genetic cause for the haplo-lethal phenotype of MAT-2 strains will be part of future research. So far we could not identify a specific SNP or allele responsible for this phenotype.

*A seemingly very low frequency of integrative transformation could be an impediment but may not have a structural basis; in Ustilago, a difference of 100-1000-fold is seen between transformation efficiencies with episomal constructs versus integrative ones. Improving transformation efficiencies will likely lead to better integration frequencies as well.*

We hope as well and work still on improvements for the transformation.

References

1) Bauch, R., Untersuchungen über die Entwicklungsgeschichte und Sexualphysiologie der Ustilago bromivora und Ustilago grandis. Zeitschrift für Botanik, 1924. XVII.

2) Vanky, K., Smut fungi of the world. 2011, Minnesota: APS Press.

3) Fisher, G.W., Holton, C. S., Biology and Control of the Smut Fungi. Vol. 1. 1957, New York: The Ronald Press Company.

4) Barbieri, M., T.C. Marcel, and R.E. Niks, Host Status of False Brome Grass to the Leaf Rust Fungus Puccinia brachypodii and the Stripe Rust Fungus P. striiformis. Plant Disease, 2011. 95(11): p. 1339-1345.

5) Barbieri, M., et al., QTLs for resistance to the false brome rust Puccinia brachypodii in the model grass Brachypodium distachyon L. Genome, 2012. 55(2): p. 152-63.

6) Bauch, R., Untersuchungen über die Entwicklungsgeschichte und Sexualphysiologie der Ustilago bromivora und Ustilago grandis. Zeitschrift für Botanik, 1924. XVII: p. 47.

7) Laurie, J.D., et al., Genome comparison of barley and maize smut fungi reveals targeted loss of RNA silencing components and species-specific presence of transposable elements. Plant Cell, 2012. 24(5): p. 1733-45.

8) Nielsen, J., Isolation and culture of monokaryotic haplonts of Ustilago nuda, the role of proline in their metabolism, and the inoculation of barley with resynthesized dikaryons. Canadian Journal of Botany, 1968. 46(10): p. 1193-1200.